# EVA: PRACTICAL SECOND-ORDER OPTIMIZATION WITH KRONECKER-VECTORIZED APPROXIMATION

**Lin Zhang**[1], **Shaohuai Shi**[2] *, **Li Bo**[1]
[1]Hong Kong University of Science and Technology, [2]Harbin Institute of Technology, Shenzhen
`lzhangbv@connect.ust.hk, shaohuais@hit.edu.cn, bli@cse.ust.hk`

## ABSTRACT

Second-order optimization algorithms exhibit excellent convergence properties for training deep learning models, but often incur significant computation and memory overheads. This can result in lower training efficiency than the first-order counterparts such as stochastic gradient descent (SGD). In this work, we present a memory- and time-efficient second-order algorithm named Eva with two novel techniques: 1) we construct the second-order information with the Kronecker factorization of small stochastic vectors over a mini-batch of training data to reduce memory consumption, and 2) we derive an efficient update formula without explicitly computing the inverse of matrices using the Sherman-Morrison formula. We further provide a theoretical interpretation of Eva from a trust-region optimization point of view to understand how it works. Extensive experimental results on different models and datasets show that Eva reduces the end-to-end training time up to $2.05\times$ and $2.42\times$ compared to first-order SGD and second-order algorithms (K-FAC and Shampoo), respectively.

## 1 INTRODUCTION

While first-order optimizers such as stochastic gradient descent (SGD) (Bottou et al., 1998) and Adam (Kingma & Ba, 2015) have been widely used in training deep learning models (Krizhevsky et al., 2012; He et al., 2016; Devlin et al., 2019), these methods require a large number of iterations to converge by exploiting only the first-order gradient to update the model parameter (Bottou et al., 2018). To overcome such inefficiency, second-order optimizers have been considered with the potential to accelerate the training process with a much fewer number of iterations to converge (Osawa et al., 2019; 2020; Pauloski et al., 2020; 2021). For example, our experimental results illustrate that second-order optimizers, e.g., K-FAC (Martens & Grosse, 2015), require ~50% fewer iterations to reach the target top-1 validation accuracy of 93.5% than SGD, in training a ResNet-110 (He et al., 2016) model on the Cifar-10 dataset (Krizhevsky, 2009) (more results are shown in Table 2).

The fast convergence property of second-order algorithms benefits from preconditioning the gradient with the inverse of a matrix $C$ of curvature information. Different second-order optimizers construct $C$ by approximating different second-order information, e.g., Hessian, Gauss-Newton, and Fisher information (Amari, 1998), to help improve the convergence rate (Dennis & Schnabel, 1983). However, classical second-order optimizers incur significant computation and memory overheads in training deep neural networks (DNNs), which typically have a large number of model parameters, as they require a quadratic memory complexity to store $C$, and a cubic time complexity to invert $C$, w.r.t. the number of model parameters. For example, a ResNet-50 (He et al., 2016) model with 25.6M parameters has to store more than 650T elements in $C$ using full Hessian, which is not affordable on current devices, e.g., an Nvidia A100 GPU has 80GB memory.

To make second-order optimizers practical in deep learning, approximation techniques have been proposed to estimate $C$ with *smaller matrices*. For example, the K-FAC algorithm (Martens & Grosse, 2015) uses the Kronecker factorization of two smaller matrices to approximate the Fisher information matrix (FIM) in each DNN layer, thus, K-FAC only needs to store and invert these small matrices, namely Kronecker factors (KFs), to reduce the computing and memory overheads.

---

*Corresponding author.

Table 1: Time and memory complexity comparison of different second-order algorithms. $d$ is the dimension of a hidden layer, $L$ is the number of layers, and $m$ is the number of gradient copies.

| Complexity | Newton | K-FAC | Shampoo | M-FAC | Eva |
|---|---|---|---|---|---|
| Time | $O(d^6 L^3)$ | $O(2d^3 L)$ | $O(2d^3 L)$ | $O(md^2 L)$ | $O(d^2 L)$ |
| Memory | $O(d^4 L^2)$ | $O(2d^2 L)$ | $O(2d^2 L)$ | $O(md^2 L)$ | $O(2dL)$ |
| Second-order Info. | Hessian | KFs | Gradient Statistics | Gradient Copies | KVs |

However, even by doing so, the additional costs of each second-order update are still significant, which makes it slower than first-order SGD. In our experiment, the iteration time of K-FAC is $2.5\times$ than that of SGD in training ResNet-50 (see Table 4), and the memory consumption of storing KFs and their inverse results is $12\times$ larger than that of storing the gradient. Despite the reduced number of iterations, existing second-order algorithms, including K-FAC (Martens & Grosse, 2015), Shampoo (Gupta et al., 2018), and M-FAC (Frantar et al., 2021), are *not* time-and-memory efficient, as shown in Table 1. One limitation in K-FAC and Shampoo is that they typically require dedicated system optimizations and second-order update interval tuning to outperform the first-order counterpart (Osawa et al., 2019; Pauloski et al., 2020; Anil et al., 2021; Shi et al., 2021).

To address the above limitations, we propose a novel second-order training algorithm, called Eva, which introduces a matrix-free approximation to the second-order matrix to precondition the gradient. Eva not only requires much less memory to estimate the second-order information, but it also does not need to explicitly compute the inverse of the second-order matrix, thus eliminating the intensive computations required in existing methods. Specifically, we propose two novel techniques in Eva. First, for each DNN layer, we exploit the Kronecker factorization of two small stochastic vectors, called Kronecker vectors (KVs), over a mini-batch of training data to construct a rank-one matrix to be the second-order matrix $C$ for preconditioning. Note that our constructed second-order matrix is different from the average outer-product of gradient (i.e., Fisher information (Amari, 1998)) that has been used in existing K-FAC related algorithms (Martens & Grosse, 2015; George et al., 2018) (§3.1). KVs require only a sublinear memory complexity w.r.t. the model size, which is much smaller than the linear memory complexity in existing second-order algorithms like storing KFs in K-FAC (Martens & Grosse, 2015), gradient statistics in Shampoo (Gupta et al., 2018), or gradient copies in M-FAC (Frantar et al., 2021) (see Table 1). Second, we derive a new update formula to precondition the gradient by implicitly computing the inverse of the constructed Kronecker factorization using the Sherman–Morrison formula (Sherman & Morrison, 1950). The new update formula takes only a linear time complexity; it means that Eva is much more time-efficient than existing second-order optimizers which normally take a superlinear time complexity in inverting matrices (see Table 1). Finally, we provide a theoretical interpretation to Eva from a trust-region optimization point of view to understand how it preserves the fast convergence property of second-order optimization (Asi & Duchi, 2019; Bae et al., 2022).

We conduct extensive experiments to illustrate the effectiveness and efficiency of Eva compared to widely used first-order (SGD, Adagrad, and Adam) and second-order (K-FAC, Shampoo, and M-FAC) optimizers on multiple deep models and datasets. The experimental results show that 1) Eva outperforms first-order optimizers – achieving higher accuracy under the same number of iterations or reaching the same accuracy with fewer number of iterations, and 2) Eva generalizes very closely to other second-order algorithms such as K-FAC while having much less iteration time and memory footprint. Specifically, in terms of per-iteration time performance, Eva only requires an average of $1.14\times$ wall-clock time over first-order SGD, while K-FAC requires $3.47\times$ in each second-order update. In term of memory consumption, Eva requires almost the same memory as first-order SGD, which is up to $31\%$ and $45\%$ smaller than second-order Shampoo and K-FAC respectively. In term of the end-to-end training performance, Eva reduces the training time on different training benchmarks up to $2.05\times$, $1.58\times$, and $2.42\times$ compared to SGD, K-FAC, and Shampoo respectively.

In summary, our contributions are as follows: (1) We propose a novel efficient second-order optimizer Eva via Kronecker-vectorized approximation, which uses the Kronecker factorization of two small vectors to be second-order information so that Eva has a sublinear memory complexity and requires almost the same memory footprint as first-order algorithms like SGD. (2) We derive a new update formula with an implicit inverse computation in preconditioning by exploiting the Sherman–Morrison formula to eliminate the expensive explicit inverse computation. Thus, Eva reduces each second-order update cost to linear time complexity. (3) We conduct extensive experiments to validate that Eva can converge faster than SGD, and it is more system efficient than K-FAC and Shampoo. Therefore, Eva is capable of improving end-to-end training performance.

## 2 BACKGROUND

In this section, we introduce the background of first-order and second-order optimization algorithms. We start from the supervised learning, in which a DNN model is trained by randomly going through the dataset $D$ many times (i.e., epochs) to minimize a loss function $\ell(\mathbf{w}, D)$. The loss function measures the average distance between model predictions and ground-truth labels, and $\mathbf{w}$ is the model parameter

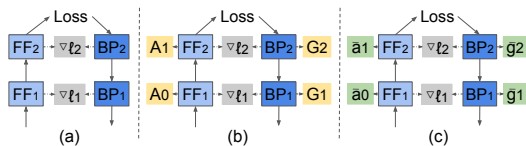

Figure 1: Examples of different optimization algorithms in a two-layer DNN model: (a) SGD, (b) K-FAC, (c) Eva.

should be trained. Given a DNN with $L$ learnable layers, the model parameter typically consists of a set of parameter matrices $\{W_l\}_{l=1}^{L}$.

**SGD.** The stochastic gradient descent (SGD) algorithm and its variants (e.g., Adam) with first-order gradient information are the main optimizers for training DNN models. In each iteration with a mini-batch of data, SGD updates the model parameter layer-wisely as follows:

$$W_l^{(t+1)} = W_l^{(t)} - \alpha^{(t)} \nabla \ell_l(\mathbf{w}^{(t)}, \mathcal{B}^{(t)}), \tag{1}$$

where $W_l^{(t)}, \nabla \ell_l(\mathbf{w}^{(t)}, \mathcal{B}^{(t)}) \in \mathbb{R}^{d_l \times d_{l-1}}$ are model parameter and first-order gradient matrices at layer $l$, respectively. $\alpha^{(t)} > 0$ is the learning rate at iteration $t$. The mini-batch of data $\mathcal{B}^{(t)}$ is sampled randomly from the training dataset. SGD typically takes a large number of iterations to converge as it only utilizes the first-order gradient to optimize the model (Bottou et al., 2018).

Compared to SGD, second-order algorithms precondition the gradient by the inverse of the curvature matrix, i.e., $C_l^{-1} \text{vec}(\nabla \ell_l)$, for model update. The straight-forward way is to use the Hessian to precondition the gradient using the Newton method, but it takes extremely high time and memory complexity as shown in Table 1. Due to the large model size in DNNs, storing and inverting $C_l$ are expensive, which motivates approximation approaches to alleviate the memory and computation complexity. One of representative methods is K-FAC (Martens & Grosse, 2015) that has been applied in large-scale training to achieve comparable performance over SGD (Osawa et al., 2019; 2020; Pauloski et al., 2020; 2021). Our proposed method in this work is also close to K-FAC (§3.1).

**K-FAC.** The K-FAC algorithm (Martens & Grosse, 2015) adopts the Fisher information matrix (FIM) as $C_l$ for layer $l$ in a DNN and approximates $C_l$ with the Kronecker product of two smaller matrices (Martens & Grosse, 2015; Grosse & Martens, 2016) [1]. For example, for a linear layer $\mathbf{a}_l = \phi(W_l \mathbf{a}_{l-1})$, in which $\phi$ is an element-wise non-linear activation function, the FIM can be approximated via $F_l = A_{l-1} \otimes G_l$, where

$$A_{l-1} = \mathbb{E}[\mathbf{a}_{l-1} \mathbf{a}_{l-1}^T] \quad \text{and} \quad G_l = \mathbb{E}[\mathbf{g}_l \mathbf{g}_l^T]. \tag{2}$$

$A_{l-1} \in \mathbb{R}^{d_{l-1} \times d_{l-1}}$ and $G_l \in \mathbb{R}^{d_l \times d_l}$ are symmetric matrices, called Kronecker factors (KFs). $\mathbf{a}_{l-1} \in \mathbb{R}^{d_{l-1}}$ is the input vector of layer $l$ (i.e., output of layer $l-1$) and $\mathbf{g}_l$ is the pre-activation gradient vector of layer $l$. $\otimes$ denotes the Kronecker product. KFs are used to precondition the gradient with a damping parameter $\gamma > 0$ via $(A_{l-1} \otimes G_l + \gamma I)^{-1} \text{vec}(\nabla \ell_l)$. Formally, we have

$$W_l^{(t+1)} = W_l^{(t)} - \alpha^{(t)} (G_l + \frac{\sqrt{\gamma}}{\pi_l} I)^{-1} \nabla \ell_l(\mathbf{w}^{(t)}, \mathcal{B}^{(t)}) (A_{l-1} + \pi_l \sqrt{\gamma} I)^{-1}, \tag{3}$$

where the scalar $\pi_l = \sqrt{T(A_{l-1})}/\sqrt{T(G_l)}$ and $T(A)$ is the trace of $A$ divided by its dimension. The detailed derivation of K-FAC is given in Appendix A.2. An example of a 2-layer DNN is shown in Fig. 1(a) and Fig. 1(b) to demonstrate the difference between SGD and K-FAC. Compared to the training process of SGD, K-FAC extra constructs two KFs, which have a memory complexity of $O(2d^2)$, to approximate FIM and computes its inverse, which has a time complexity of $O(2d^3)$, to precondition the gradient in each layer.

**Additional approaches.** There exist some other second-order optimizers for DNN training. For example, Shampoo (Gupta et al., 2018) is a full-matrix adaptive algorithm that constructs gradient statistics matrices (similar to KFs) by building smaller matrices layer-wisely for preconditioning,

---

[1]In this paper, we focus on the oft-used K-FAC with the *empirical* FIM, as discussed in Appendix A.2.

which introduces a time complexity of $O(2d^3L)$ to invert these matrices, and a memory complexity of $O(2d^2L)$ to store them. $d$ is the dimension of a hidden layer. M-FAC (Frantar et al., 2021) is a matrix-free algorithm that utilizes matrix-vector products with many gradient copies. M-FAC does not invert any matrix but requires $m$ copies of gradient to estimate FIM, which brings time and memory costs of $O(md^2L)$. $m$ is typically suggested to be as high as 1024 (Frantar et al., 2021).

In summary, existing second-order methods have high time and/or memory complexity in each second-order update as shown in Table 1. In this work, we present a novel second-order optimizer, Eva, with much lower time and memory complexity.

## 3 EVA: KRONECKER-FACTORED APPROXIMATION WITH SMALL VECTORS

Following standard feed-forward (FF) and back-propagation (BP) processes, as shown in Fig. 1(c), we propose to construct the second-order curvature matrix, denoted as $R_l$ for the sake of clarity, for layer $l$ using the Kronecker product of two small vectors $\bar{\mathbf{a}}_{l-1}$ and $\bar{\mathbf{g}}_l$, i.e.,

$$R_l = (\bar{\mathbf{a}}_{l-1}\bar{\mathbf{a}}_{l-1}^T) \otimes (\bar{\mathbf{g}}_l\bar{\mathbf{g}}_l^T) = (\bar{\mathbf{a}}_{l-1} \otimes \bar{\mathbf{g}}_l)(\bar{\mathbf{a}}_{l-1} \otimes \bar{\mathbf{g}}_l)^T, \tag{4}$$

where

$$\bar{\mathbf{a}}_{l-1} = \mathbb{E}[\mathbf{a}_{l-1}] = \frac{1}{|\mathcal{B}|}\sum_{i\in\mathcal{B}}\mathbf{a}_{l-1}^{(i)} \quad \text{and} \quad \bar{\mathbf{g}}_l = \mathbb{E}[\mathbf{g}_l] = \frac{1}{|\mathcal{B}|}\sum_{i\in\mathcal{B}}\mathbf{g}_l^{(i)}. \tag{5}$$

$\mathbf{a}_{l-1}^{(i)}$ and $\mathbf{g}_l^{(i)}$ are activation and pre-activation gradient vectors at layer $l$, respectively, sampled from a mini-batch of data $\mathcal{B}$. We call $\bar{\mathbf{a}}_{l-1}$ and $\bar{\mathbf{g}}_l$ as Kronecker vectors (KVs), whose dimensions are the same with $\mathbf{a}_{l-1}^{(i)}$ and $\mathbf{g}_l^{(i)}$ respectively and they are irrelevant with the batch size. Thus, we just need to store inexpensive KVs to construct $R_l$, which is much more memory-efficient than storing the whole matrix like K-FAC or Shampoo. The preconditioner of Eva becomes $(R_l + \gamma I)^{-1}$. However, it still needs to compute the inverse of the damped $R_l$, which is computation-expensive in training DNNs. From Eq. 4, $R_l$ is a rank-one matrix. Using this good property, we can implicitly compute its inverse efficiently using the Sherman-Morrison formula (Sherman & Morrison, 1950). Specifically,

$$(R_l + \gamma I)^{-1} = ((\bar{\mathbf{a}}_{l-1} \otimes \bar{\mathbf{g}}_l)(\bar{\mathbf{a}}_{l-1} \otimes \bar{\mathbf{g}}_l)^T + \gamma I)^{-1} = \frac{1}{\gamma}(I - \frac{(\bar{\mathbf{a}}_{l-1}\bar{\mathbf{a}}_{l-1}^T) \otimes (\bar{\mathbf{g}}_l\bar{\mathbf{g}}_l^T)}{(\bar{\mathbf{a}}_{l-1}^T\bar{\mathbf{a}}_{l-1})(\bar{\mathbf{g}}_l^T\bar{\mathbf{g}}_l) + \gamma}). \tag{6}$$

Based on the above preconditioner, we can derive the update formula of Eva to be

$$W_l^{(t+1)} = W_l^{(t)} - \frac{\alpha^{(t)}}{\gamma}\left(\nabla\ell_l(\mathbf{w}^{(t)}, \mathcal{B}^{(t)}) - \frac{\bar{\mathbf{g}}_l^T\nabla\ell_l(\mathbf{w}^{(t)}, \mathcal{B}^{(t)})\bar{\mathbf{a}}_{l-1}}{(\bar{\mathbf{a}}_{l-1}^T\bar{\mathbf{a}}_{l-1})(\bar{\mathbf{g}}_l^T\bar{\mathbf{g}}_l) + \gamma}\bar{\mathbf{g}}_l\bar{\mathbf{a}}_{l-1}^T\right). \tag{7}$$

The detailed derivation of Eva is deferred to Appendix B.2 and the study of its training dynamics is provided in Fig. 6 in Appendix C. Compared to SGD, the precondition process of Eva changes the gradient direction with compensating in the direction of $\bar{\mathbf{g}}_l\bar{\mathbf{a}}_{l-1}^T$ and scales the step size by $1/\gamma$.

In Eva, KVs are calculated over a mini-batch of data for constructing the second-order preconditioner by their Kronecker product. For a large-scale training dataset, the approximation can be stabilized by a long-term run of estimation using the whole data. Thus, we use the running average strategy which is also commonly used in Adam Kingma & Ba (2015) with 1st and 2rd moments, or K-FAC (Martens & Grosse, 2015) with KFs, that is

$$\bar{\mathbf{a}}_{l-1}^{(t)} \leftarrow \xi\bar{\mathbf{a}}_{l-1}^{(t)} + (1-\xi)\bar{\mathbf{a}}_{l-1}^{(t-1)} \quad \text{and} \quad \bar{\mathbf{g}}_l^{(t)} \leftarrow \xi\bar{\mathbf{g}}_l^{(t)} + (1-\xi)\bar{\mathbf{g}}_l^{(t-1)}, \tag{8}$$

where $\xi \in (0, 1]$ is the running average parameter in iteration $t$, $\bar{\mathbf{a}}_{l-1}^{(t)}$ and $\bar{\mathbf{g}}_l^{(t)}$ are new KVs calculated during each FF and BP computation and they are used to update the state of KVs.

**Complexity analysis.** The extra time costs of Eva come from constructing KVs and preconditioning the gradients with KVs, apart from sharing the same FF, BP, and update computations as SGD. As the overhead of estimating KVs can be ignored, the main time cost of Eva is multiple vector multiplications for preconditioning, which can be denoted as $O(d^2L)$. It means Eva has a time complexity that is *linear* to the total number of parameters, which is much smaller than the superlinear complexity in K-FAC and Shampoo as shown in Table 1. Besides, the memory complexity of Eva is $O(2dL)$ for storing KVs, which is *sublinear* to the total number of parameters. In summary, Eva has very little extra time and memory costs in each second-order update compared to first-order SGD, but it enjoys the fast convergence property of second-order K-FAC.

## 3.1 THEORETICAL UNDERSTANDING

**Trust-region optimization.** The trust-region optimization algorithm (Yuan, 2015) can be formulated as

$$\mathbf{w}^{(t+1)} = \min_{\mathbf{w}} \ell(\mathbf{w}), \qquad (9)$$

$$\text{s.t. } \rho(\mathbf{w}, \mathbf{w}^{(t)}) \leq \lambda, \qquad (10)$$

where $\mathbf{w}^{(t)}$ and $\mathbf{w}^{(t+1)}$ are current and next parameter points, respectively. $\rho(\mathbf{w}, \mathbf{w}^{(t)})$ is the

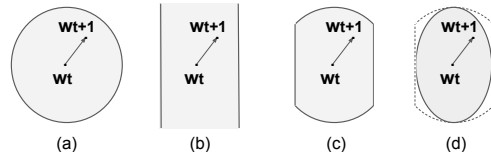

Figure 2: Trust regions of different proximal constraints: (a) ball region, (b) strip region, (c) trust region of Eva, (d) trust region of K-FAC.

proximal function to measure the distance between $\mathbf{w}$ and $\mathbf{w}^{(t)}$ (the smaller the closer). Trust-region optimization treats each update step as finding the next parameter point that minimizes the loss function and is close to the current parameter point (i.e., located in a trust region of $\{\mathbf{w} : \rho(\mathbf{w}, \mathbf{w}^{(t)}) \leq \lambda\}$, where $\lambda > 0$ is the threshold). Different proximal functions correspond to different trust regions. For example, the euclidean distance, $\rho(\mathbf{w}, \mathbf{w}^{(t)}) = ||\mathbf{w} - \mathbf{w}^{(t)}||^2$, gives a ball region as shown in Fig. 2(a).

Adopting the same assumption of K-FAC (Martens & Grosse, 2015), where the input and pre-activation gradient vectors are fairly independent, i.e., $\mathbb{E}[\mathbf{a} \otimes \mathbf{g}] \approx \mathbb{E}[\mathbf{a}] \otimes \mathbb{E}[\mathbf{g}]$, we have

$$R = (\mathbb{E}[\mathbf{a}] \otimes \mathbb{E}[\mathbf{g}])(\mathbb{E}[\mathbf{a}] \otimes \mathbb{E}[\mathbf{g}])^T \approx \mathbb{E}[\mathbf{a} \otimes \mathbf{g}]\mathbb{E}[\mathbf{a} \otimes \mathbf{g}]^T = \mathbb{E}[\text{vec}(\nabla \ell)]\mathbb{E}[\text{vec}(\nabla \ell)]^T, \quad (11)$$

where $\mathbb{E}[\text{vec}(\nabla \ell)]$ is the estimated gradient vector over the current mini-batch of data. The above equation indicates that the second-order matrix $R$ in Eva is in fact the approximation of the outer product of average gradient (OPAG) (i.e., $\mathbb{E}[\text{vec}(\nabla \ell)]\mathbb{E}[\text{vec}(\nabla \ell)]^T$). Here we ignore the subscription of the layer index for simplicity. By applying OPAG into the proximal function to measure the parameter closeness, we obtain

$$\rho(\mathbf{w}, \mathbf{w}^{(t)}) = (\mathbf{w} - \mathbf{w}^{(t)})^T R(\mathbf{w} - \mathbf{w}^{(t)}) = (\Delta \mathbf{w}^T \mathbb{E}[\text{vec}(\nabla \ell)])^2 \leq \lambda, \qquad (12)$$

where $\Delta \mathbf{w} = \mathbf{w} - \mathbf{w}^{(t)}$ is the parameter update. Therefore, the corresponding trust region is as shown in Fig. 2(b), which limits the update size along the gradient direction. According to Eq. 7, Eva use the preconditioned gradient using the damped OPAG as the parameter update, i.e., $\Delta \mathbf{w} = (R + \gamma I)^{-1}\text{vec}(\nabla \ell)$. It indicates that the trust region of Eva is an intersection between the ball region and the strip region, as shown in Fig. 2(c). More details are given in Appendix B.

**Relation to K-FAC.** From the trust-region optimization perspective, the K-FAC algorithm with the *true* FIM is known to choose $\rho$ as KL-divergence between model distributions (Bae et al., 2022), which however is not the case for the K-FAC with *empirical* FIM that has been widely applied in large-scale training (Osawa et al., 2019; 2020; Ueno et al., 2020; Pauloski et al., 2020; 2021). We focus on the analysis on oft-used K-FAC with empirical FIM as it is much more practical than true FIM. We build the connection between K-FAC with empirical FIM (Osawa et al., 2019; 2020) and our proposed Eva. Formally, the relationship between the empirical FIM and OPAG is:

$$\underbrace{\mathbb{E}[\text{vec}(\nabla \ell)\text{vec}(\nabla \ell)^T]}_{\text{empirical FIM}} = \underbrace{\mathbb{E}[\text{vec}(\nabla \ell)]\mathbb{E}[\text{vec}(\nabla \ell)]^T}_{\text{OPAG: outer-product of average gradient}} + \underbrace{\text{Cov}(\text{vec}(\nabla \ell))}_{\text{gradient covariance matrix}}, \qquad (13)$$

which indicates the empirical FIM is the sum of OPAG and the gradient covariance matrix (McCandlish et al., 2018). K-FAC and Eva utilize the approximate empirical FIM and approximate OPAG as second-order information for calculating preconditioners, respectively. That is $F \approx \mathbb{E}[\text{vec}(\nabla \ell)\text{vec}(\nabla \ell)^T]$ and $R \approx \mathbb{E}[\text{vec}(\nabla \ell)]\mathbb{E}[\text{vec}(\nabla \ell)]^T$. As $\text{Cov}(\text{vec}(\nabla \ell))$ in Eq. 13 is positive semi-definite, we have $\mathbf{v}^T F \mathbf{v} \geq \mathbf{v}^T R \mathbf{v}$ for any vector $\mathbf{v}$. Thus, we have

$$\{\mathbf{w} : \Delta \mathbf{w}^T (F + \gamma I)\Delta \mathbf{w} \leq \lambda\} \subseteq \{\mathbf{w} : \Delta \mathbf{w}^T (R + \gamma I)\Delta \mathbf{w} \leq \lambda\}, \qquad (14)$$

for a given threshold $\lambda$, which indicates the trust region of Eva is larger than K-FAC as shown in Fig. 2(d). Therefore, the update step of K-FAC to minimize the loss is more conservative than Eva.

## 4 EVALUATION

### 4.1 CONVERGENCE PERFORMANCE

We evaluate the generalization performance of Eva with three representative models, VGG-19 (Simonyan & Zisserman, 2015), ResNet-110 (He et al., 2016), and WideResNet-28-10 (WRN-28-

Table 2: Validation accuracy (%) comparison between Eva and SGD/K-FAC algorithms for training from scratch with different epoch buckets. † indicates training with extra tricks.

| Model | Epoch | Cifar-10 | | | Cifar-100 | | |
|---|---|---|---|---|---|---|---|
| | | SGD | K-FAC | Eva | SGD | K-FAC | Eva |
| VGG-19 | 50 | $90.95_{\pm0.2}$ | $92.57_{\pm0.3}$ | $\mathbf{92.63}_{\pm0.2}$ | $61.69_{\pm0.8}$ | $70.14_{\pm0.1}$ | $\mathbf{70.23}_{\pm0.7}$ |
| | 100 | $92.27_{\pm0.3}$ | $\mathbf{93.37}_{\pm0.2}$ | $93.20_{\pm0.1}$ | $67.97_{\pm0.3}$ | $\mathbf{72.25}_{\pm0.1}$ | $71.79_{\pm0.5}$ |
| | 200 | $93.02_{\pm0.1}$ | $93.46_{\pm0.2}$ | $\mathbf{93.59}_{\pm0.2}$ | $70.98_{\pm0.0}$ | $\mathbf{72.90}_{\pm0.3}$ | $72.72_{\pm0.4}$ |
| ResNet-110 | 50 | $90.98_{\pm0.6}$ | $\mathbf{93.03}_{\pm0.1}$ | $93.02_{\pm0.3}$ | $67.93_{\pm0.9}$ | $71.03_{\pm0.5}$ | $\mathbf{71.13}_{\pm0.4}$ |
| | 100 | $92.49_{\pm0.6}$ | $93.76_{\pm0.1}$ | $93.76_{\pm0.0}$ | $70.74_{\pm0.7}$ | $72.31_{\pm0.5}$ | $\mathbf{72.38}_{\pm0.2}$ |
| | 200 | $93.80_{\pm0.2}$ | $\mathbf{94.21}_{\pm0.2}$ | $93.99_{\pm0.1}$ | $72.43_{\pm0.6}$ | $72.96_{\pm0.3}$ | $\mathbf{73.29}_{\pm0.3}$ |
| WRN-28-10† | 50 | $95.28_{\pm0.1}$ | $96.12_{\pm0.2}$ | $\mathbf{96.19}_{\pm0.1}$ | $79.03_{\pm0.1}$ | $80.98_{\pm0.2}$ | $\mathbf{81.15}_{\pm0.2}$ |
| | 100 | $96.88_{\pm0.2}$ | $\mathbf{97.21}_{\pm0.0}$ | $97.05_{\pm0.0}$ | $83.03_{\pm0.3}$ | $\mathbf{83.54}_{\pm0.4}$ | $83.52_{\pm0.2}$ |
| | 200 | $97.33_{\pm0.1}$ | $\mathbf{97.44}_{\pm0.1}$ | $97.38_{\pm0.1}$ | $84.54_{\pm0.0}$ | $\mathbf{84.56}_{\pm0.1}$ | $84.52_{\pm0.1}$ |

Table 3: Validation accuracy (%) comparison between Eva and 4 more algorithms for training from scratch on Cifar-10 with 100 epochs. † indicates training with extra tricks (except M-FAC).

| Model | Adagrad | AdamW | Shampoo | M-FAC | Eva |
|---|---|---|---|---|---|
| VGG-19 | $92.42_{\pm0.0}$ | $92.97_{\pm0.1}$ | $\mathbf{93.38}_{\pm0.1}$ | $92.50_{\pm0.1}$ | $93.20_{\pm0.1}$ |
| ResNet-110 | $90.34_{\pm0.2}$ | $92.61_{\pm0.1}$ | $92.47_{\pm0.2}$ | $93.45_{\pm0.1}$ | $\mathbf{93.76}_{\pm0.0}$ |
| WRN-28-10† | $93.72_{\pm0.2}$ | $96.91_{\pm0.0}$ | $96.99_{\pm0.1}$ | $94.54_{\pm0.2}$ | $\mathbf{97.05}_{\pm0.0}$ |

10) (Zagoruyko & Komodakis, 2016) on Cifar-10 (Krizhevsky, 2009) and Cifar-100 (Krizhevsky, 2009) datasets. We compare Eva to the first-order baseline SGD (with a momentum of 0.9), and the second-order baseline K-FAC (Martens & Grosse, 2015). Following the configurations of (Pauloski et al., 2020; 2021), we set the same hyper-parameters for all algorithms for a fair comparison, and the details are given in Appendix C.1. Since training with more epochs can generalize better (Hoffer et al., 2017), we run each algorithm with 50, 100, and 200 epochs for a better comparison from compressed to sufficient training budgets. The validation accuracy comparison is shown in Table 2 and Table 3, where we report the mean and std over three independent runs. We summarize the results in the following three aspects.

First, in the models VGG-19 and ResNet-110 with commonly used settings from their original papers (details in Appendix C.1), it is seen that Eva performs closely to the second-order K-FAC, and they both consistently outperform SGD under different training budgets. To be specific, under the 50-epoch setting, both Eva and K-FAC outperform SGD by a large margin, e.g., $+8.5\%$ for training VGG-19 on Cifar-100; under the setting of sufficient 200 epochs, Eva and K-FAC still achieve slightly better generalization performance than SGD. Due to the page limit, we put the convergence curves under the 50-epoch setting in Fig. 7 in Appendix C , which shows that Eva converges similarly as K-FAC and learns much faster than SGD. Eva and K-FAC with 50 (and 100) epochs achieve the loss or validation accuracy that SGD needs to take 100 (and 200) epochs. In summary, our proposed second-order Eva has similar good convergence performance with K-FAC, and both of them train the models around $2\times$ faster than SGD in terms of iterations. It validates that second-order algorithms have better convergence performance than SGD and they can reach the same target accuracy in fewer number of training iterations.

Second, considering that existing training paradigms typically use extra tricks like CutMix (Yun et al., 2019) and AutoAugment (Cubuk et al., 2019) to achieve better validation accuracy, we train a relatively new model WRN-28-10 (Zagoruyko & Komodakis, 2016) on Cifar-10 and Cifar-100 to compare the performance of different optimizers. Note CutMix (Yun et al., 2019) and AutoAugment (Cubuk et al., 2019) have been particularly developed and heavily tuned for the first-order SGD optimizer, which could be detrimental to second-order algorithms. However, as shown in Table 2, our Eva and K-FAC still learn faster and generalize better than SGD under the same number of training epochs. Additional results to verify the generalization performance for fine-tuning pretrained models are provided in Table 6 in Appendix C. In summary, Eva achieves the same generalization performance with a less number of iterations than well-tuned SGD or it achieves higher generalization performance with the same training number of iterations as SGD.

Third, we compare the convergence performance of Eva with 4 more popular optimizers: Adagrad (Duchi et al., 2010), AdamW (Loshchilov & Hutter, 2019), Shampoo (Gupta et al., 2018), and M-FAC (Frantar et al., 2021). Adagrad and AdamW are common adaptive gradient methods, while

Shampoo and M-FAC are recently proposed second-order algorithms. We tune the learning rate for each algorithm to choose a best one for particular algorithms (see Appendix C.1) to train three models on Cifar-10 for 100 epochs. The results are given in Table 3, showing that Eva achieves comparable performance to other second-order algorithms Shampoo and M-FAC, and outperforms first-order adaptive methods Adagrad and AdamW on different models. M-FAC, however, results in 2.5% accuracy loss, compared to Eva and Shampoo in training WRN-28-10. Note that we exclude CutMix and AutoAugment in M-FAC as they cause M-FAC divergences in training WRN-28-10.

Table 4: Relative iteration time and memory over SGD. Values in parentheses represent the results with increased second-order update intervals (10 on Cifar-10 and 50 on ImageNet).

| Dataset | Model | Shampoo | | K-FAC | | Eva | |
|---------|-------|---------|-----|-------|-----|-----|-----|
| | | Time | Mem | Time | Mem | Time | Mem |
| | VGG-19 | $19.6\times$ $(2.89\times)$ | $1.01\times$ | $5.57\times$ $(1.68\times)$ | $1.01\times$ | **$1.13\times$** | **$1.00\times$** |
| Cifar-10 | ResNet-110 | $6.79\times$ $(1.90\times)$ | $1.00\times$ | $1.64\times$ $(1.16\times)$ | $1.03\times$ | **$1.16\times$** | **$1.00\times$** |
| | WRN-28-10 | $6.69\times$ $(1.60\times)$ | $1.05\times$ | $2.68\times$ $(1.25\times)$ | $1.38\times$ | **$1.03\times$** | **$1.00\times$** |
| | ResNet-50 | $30.7\times$ $(1.71\times)$ | $1.07\times$ | $2.52\times$ $(1.14\times)$ | $1.06\times$ | **$1.09\times$** | **$1.00\times$** |
| ImageNet | Inception-v4 | $75.3\times$ $(2.70\times)$ | $1.11\times$ | $3.95\times$ $(1.28\times)$ | $1.42\times$ | **$1.28\times$** | **$1.00\times$** |
| | ViT-B/16 | $199.\times$ $(6.20\times)$ | $1.31\times$ | $4.47\times$ $(1.43\times)$ | $1.45\times$ | **$1.18\times$** | **$1.00\times$** |

## 4.2 TIME AND MEMORY EFFICIENCY

As we have shown the good convergence and generalization performance of Eva, we demonstrate its per-iteration training time and memory consumption compared with SGD, Shampoo, and K-FAC. We do not report M-FAC here as it needs to store extra $m$ gradients for FIM estimation ($m = 1024$ by default), which is very memory inefficient (normally causes out-of-memory). To cover relatively large models, we select three extra popular deep models trained on ImageNet (Deng et al., 2009): ResNet-50, Inception-v4 (Szegedy et al., 2017), and ViT-B/16 (Dosovitskiy et al., 2021). We run each algorithm on an Nvidia RTX2080Ti GPU. Following the training recipes in (Pauloski et al., 2020), we set the second-order update interval as 10 on Cifar-10, and 50 on ImageNet for both Shampoo and K-FAC to reduce their average iteration time. More detailed configurations can be found in Appendix C.1. We report relative time and memory costs over SGD in Table 4.

**Time efficiency.** We can see that Eva has the shortest iteration time among all evaluated second-order algorithms. Eva has only $1.14\times$ longer iteration time on average than SGD, but it achieves an average of $3.04\times$ and $49.3\times$ faster than K-FAC and Shampoo, respectively. This is because our Eva does not need to explicitly compute the inverse of second-order matrix while K-FAC requires expensive inverse matrix computations on KFs and Shampoo performs even more computations of inverse $p$-th roots ($p \geq 2$). Therefore, Eva is able to update second-order information iteratively to achieve faster convergence, but K-FAC and Shampoo have to update their preconditioners infrequently. For example, the average time can be reduced to $1.32\times$ and $2.83\times$ for K-FAC and Shampoo, respectively, when increasing the second-order update interval (10 on Cifar-10 and 50 on ImageNet). We will show the end-to-end training performance in §4.3.

**Memory efficiency.** In terms of memory consumption, Eva takes almost the same memory as SGD, and spends much less memory than Shampoo (with $1.09\times$) and K-FAC (with $1.22\times$), since Eva only needs to store small vectors, but Shampoo and K-FAC have to store second-order matrices and their inverse results. We notice that K-FAC consumes more GPU memory than Shampoo as K-FAC requires extra memory on intermediate states to estimate KFs, such as unfolding images.

In summary, Eva is a much more efficient second-order optimizer than K-FAC and Shampoo in terms of the average iteration time and memory consumption.

## 4.3 END-TO-END TRAINING PERFORMANCE

We further compare the end-to-end wall-clock time to reach the target accuracy training with different optimizers. We train VGG-19, ResNet-110, and WRN-28-10 on Cifar-10 with one RTX2080Ti GPU (11GB memory), and ResNet-50 on ImageNet with 32 RTX2080Ti GPUs. We set the second-order information update interval to 10 on Cifar-10 and 50 on ImageNet for K-FAC and Shampoo. More configurations can be found in Appendix C.1.

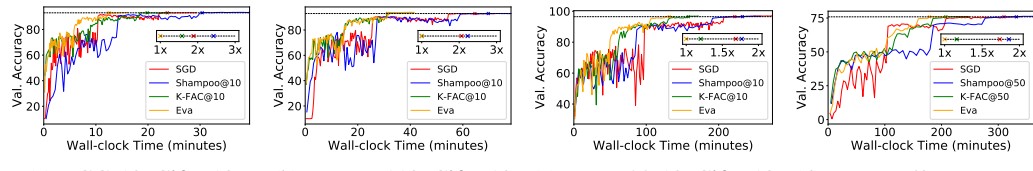

(a) VGG-19, Cifar-10    (b) ResNet-110, Cifar-10    (c) WRN-28-10, Cifar-10    (d) ResNet-50, ImageNet

Figure 3: Wall-clock time comparison between Eva and SGD/K-FAC/Shampoo algorithms for training multiple DNNs. The inset plot reports relative time-to-solution over Eva.

The results are given in Fig. 3, showing that Eva optimizes faster than all the other evaluated algorithms SGD, K-FAC, and Shampoo. On the Cifar-10 dataset, Eva is $1.88\times$, $1.26\times$, and $2.14\times$ faster on average than SGD, K-FAC, and Shampoo, respectively. This is because Eva requires less training epochs than SGD for convergence, and the iteration time of Eva is smaller than K-FAC and Shampoo. We notice that K-FAC is also possible to optimize faster than SGD, but it needs to increase the second-order update interval. Otherwise, the training time of K-FAC would be much more expensive as studied in Fig. 8 in Appendix C.

Though increasing the interval makes K-FAC and Shampoo computationally efficient, they require more GPU memory than Eva. For a fair comparison, on the large-scale ImageNet dataset, we set per-GPU batch size to 96 for SGD and Eva, and 64 for K-FAC and Shampoo, to maximize the GPU utilization. In this setting, the results shown in Fig. 3(d) indicates that Eva achieves $1.74\times$, $1.16\times$, $1.86\times$ speedups over SGD, K-FAC, and Shampoo, respectively, to achieve the target accuracy of 75.9% on the validation set according to MLPerf. The median test accuracy of the final 5 epochs on ImageNet is 76.02%, 76.25%, 76.06%, and 75.96% for SGD, Shampoo, K-FAC, and Eva, with 100, 60, 55, and 55 epochs, respectively in training ResNet-50. The throughput improvement of Eva by using larger per-GPU batch size is studied in Table 7 in Appendix C.

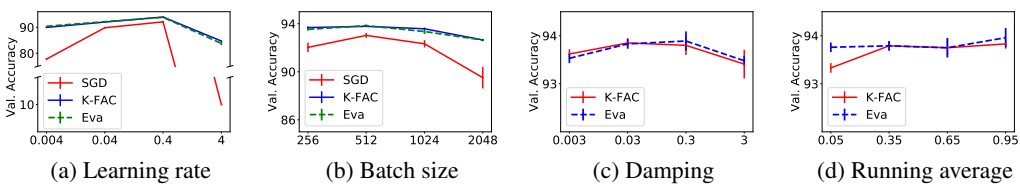

(a) Learning rate     (b) Batch size     (c) Damping     (d) Running average

Figure 4: Hyper-parameter study of Eva by training ResNet-110 on Cifar-10 with 100 epochs.

## 4.4 HYPER-PARAMETER AND ABLATION STUDY

We study the convergence performance of Eva with different hyper-parameters, including learning rate, batch size, damping, and running average. The results are shown in Fig. 4 in training ResNet-110 on Cifar-10, and the similar results of training VGG-19 on Cifar-100 are reported in Fig. 9 in Appendix C. First, the learning rate and batch size are two key hyper-parameters to both SGD and Eva, as shown in Fig. 4(a) and (b), but Eva and K-FAC can consistently outperform SGD under different settings, and largely outperform SGD in the large batch size regime. However, a large learning rate would cause performance degradation in SGD and Eva, but Eva is more robust to the learning rate and batch size than SGD. Second, we tune the damping and running average hyper-parameters introduced in K-FAC and Eva, as shown in Fig. 4(c) and (d). K-FAC and Eva perform very closely and they are both robust to damping and running average values, which implies that one can use default hyper-parameters to achieve good performance. Therefore, the hyper-parameters used in SGD and some dedicated hyper-parameters used in K-FAC can be directly applied in Eva to achieve good convergence performance instead of tuning them. Finally, we conduct the ablation study on Eva to validate the necessity of using momentum, KL clip (details in Appendix B.2), and KVs. We compare the performance of Eva to three variants without using momentum, KL clip (see Eq. 45), and KVs (see Eq. 38), respectively. The results are demonstrated in Table 5, which show that discarding any part of them would cause performance degradation. Specifically, momentum can improve the generalization of Eva (similar to SGD with momentum). KL clip is of importance

Table 5: Ablation study on Eva without using momentum, KL clip, and KVs, respectively.

| Dataset | Model | Eva | w/o momentum | w/o KL clip | w/o KVs |
|---------|-------|-----|--------------|-------------|---------|
| Cifar-10 | ResNet-110 | $\textbf{93.86}_{\pm 0.1}$ | $89.39_{\pm 0.1}$ | $90.95_{\pm 0.5}$ | $92.62_{\pm 0.2}$ |
| Cifar-10 | WRN-28-10$^\dagger$ | $\textbf{97.03}_{\pm 0.1}$ | $94.59_{\pm 0.1}$ | $67.27_{\pm 50.}$ | $96.67_{\pm 0.0}$ |
| Cifar-100 | VGG-19 | $\textbf{72.04}_{\pm 0.3}$ | $66.00_{\pm 0.3}$ | $60.93_{\pm 2.3}$ | $66.64_{\pm 0.8}$ |

to prevent exploding the preconditioned gradients (similar to K-FAC), otherwise, Eva w/o KL clip could cause divergence as shown in training WRN-28-10 on Cifar-10. In addition, KVs are required, rather than gradient norm, to construct useful curvature information that helps optimization.

### 4.5 LIMITATION AND FUTURE WORK

Though we have demonstrated the good performance of our proposed Eva, we would like to discuss several limitations and possible future work: (1) there lacks a solid theoretical analysis on the convergence rate for both Eva and K-FAC with empirical FIM; (2) since most training tricks were initially proposed for first-order algorithms, it is of interest to design novel second-order friendly strategies for achieving possibly better performance; (3) we will conduct more experiments to show the effectiveness of Eva on other applications like NLP; (4) our prototype implementation of Eva currently only supports data parallelism for distributed training, which can be further integrated with model parallelism (Huang et al., 2019; Shoeybi et al., 2019) for training very large models.

## 5 RELATED WORK

**Matrix-free methods** do not explicitly construct second-order matrix, but they rely on the matrix-vector products to calculate the preconditioned gradients. The very initial work in this line is the Hessian-free method (Martens et al., 2010), which requires only Hessian-vector products with an iterative conjugate gradient (CG) approach. To reduce the cost of each CG iteration, subsampled mini-batch can be used for Hessian-vector products (Erdogdu & Montanari, 2015). Recently, M-FAC (Frantar et al., 2021) is proposed to estimate inverse-Hessian vector products with a recursive Woodbury-Sherman-Morrison formula (Amari, 1998). However, matrix-free methods forgo second-order matrix at a cost of either performing expensive CG iterations or storing extra sliding gradients.

**Approximation methods**, on the other hand, construct smaller second-order matrix with different approximation techniques such as quasi-Newton (Goldfarb et al., 2020), quantization (Alimisis et al., 2021), Hessian diagonal (Yao et al., 2021), and the most relevant one K-FAC (Martens & Grosse, 2015; Grosse & Martens, 2016), among which K-FAC is a relatively practical one for deep learning. With K-FAC, one only needs to construct and invert KFs for preconditioning, which is much more efficient than inverting large FIM directly. However, inverting and storing KFs are not cheap enough compared with SGD, thus, many recent works attempt to accelerate K-FAC with distributed training (Osawa et al., 2019; 2020; Ueno et al., 2020; Pauloski et al., 2020; 2021; Shi et al., 2021; Zhang et al., 2022; 2023). Besides, full-matrix adaptive methods such as Shampoo (Gupta et al., 2018; Anil et al., 2021) are very similar to K-FAC, which construct second-order preconditioners for gradient tensors on each dimension (concretely, a variant of full-matrix Adagrad). However, the required inverse $p$-th root computations are more expensive than inverting KFs. These expensive memory and compute costs make them even slower than first-order SGD.

## 6 CONCLUSION

We proposed an efficient second-order algorithm called Eva to accelerate DNN training. We first proposed to use the Kronecker factorization of two small vectors to construct second-order information, which significantly reduces memory consumption. Then we derived a computational-friendly update formula without explicitly calculating the inverse of the second-order matrix using the Sherman–Morrison formula, which reduces the per-iteration computing time. We also provided a theoretical interpretation to Eva with a trust-region optimization perspective. Extensive experiments were conducted to validate its effectiveness and efficiency, and the results show that Eva outperforms existing popular first-order and second-order algorithms on multiple models and datasets.

ACKNOWLEDGMENTS

The research was supported in part by a RGC RIF grant under the contract R6021-20, and RGC GRF grants under the contracts 16209120 and 16200221.

REPRODUCIBILITY

We give precise statements for our algorithm in Section 3, and provide all the necessary implementation details in Appendix B.2. We have specified the settings and hyper-parameters required to reproduce our experimental results in Appendix C.1. We implement our algorithm atop PyTorch framework and provide easy-to-use APIs so that users can adopt it by adding several lines of code in their training scripts. The code is available at `https://github.com/lzhangbv/eva`.

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

## A    APPENDIX: BACKGROUND OF SECOND-ORDER OPTIMIZATION

### A.1    SECOND-ORDER OPTIMIZATION

Second-order optimization algorithms typically utilize the second-order information matrix $C$ to precondition the first-order gradient (i.e., multiplying the gradient by the inverse of $C$), and then use the preconditioned gradient to update the model parameter as follows:

$$\mathbf{w}^{(t+1)} = \mathbf{w}^{(t)} - \alpha^{(t)} C^{-1} \mathtt{vec}(\nabla \ell(\mathbf{w}^{(t)}, \mathcal{B}^{(t)})), \tag{15}$$

where $\mathbf{w}^{(t)}, \mathtt{vec}(\nabla \ell(\mathbf{w}^{(t)}, \mathcal{B}^{(t)})) \in \mathbb{R}^n$ are the model parameter vector, and the first-order gradient vector of a loss with respect to model parameter, respectively. $C \in \mathbb{R}^{n \times n}$ is the second-order curvature information matrix, and $\alpha^{(t)} > 0$ is the learning rate at iteration $t$. The mini-batch of data $\mathcal{B}^{(t)}$ is sampled randomly from the training dataset. The matrix-vector multiplication of $C^{-1}\mathtt{vec}(\nabla \ell)$ is called *preconditioning*, and the second-order information matrix includes Hessian, Gauss-Newton, Fisher information, their damped versions and so on (Bottou et al., 2018). When $C = I$, the update formula becomes the first-order SGD algorithm.

### A.2    DETAILED ANALYSIS OF K-FAC

K-FAC has been proven to be one of the most powerful second-order optimizers for training deep models (Osawa et al., 2019; 2020; Pauloski et al., 2020; 2021), and the idea of Kronecker factorized approximation of curvature information is similar to our algorithm. Here we provide a detailed analysis of K-FAC (Martens & Grosse, 2015), which could help us understand Eva provided in the next section (§ B.2). We start from introducing FIM, which is the curvature matrix used in K-FAC, and then establish the Kronecker factorized approximation for FIM, and give the final update formula with the damping technique.

**FIM.** In the natural gradient descent (NGD) algorithm (Amari, 1998; Martens & Grosse, 2015), the Fisher information matrix (FIM) is used and inverted to precondition the gradient. The update formula of NGD is:

$$\mathbf{w}^{(t+1)} = \mathbf{w}^{(t)} - \alpha^{(t)} F^{-1} \mathtt{vec}(\nabla \ell(\mathbf{w}^{(t)}, \mathcal{B}^{(t)})), \tag{16}$$

where $F \in \mathbb{R}^{n \times n}$ is the FIM, and $F^{-1}\mathtt{vec}(\nabla \ell)$ is the natural gradient, which is known to be the steepest direction in the space of model distributions (Amari, 1998). The true FIM is derived from

$$F_{1mc} = \mathbb{E}_{x \sim p_{data}, \hat{y} \sim r(\cdot|x)}[\mathtt{vec}(\nabla \ell)\mathtt{vec}(\nabla \ell)^T], \tag{17}$$

where $\nabla \ell = \nabla \log r(\hat{y}|x)$ is the *pseudo-gradient* w.r.t. model parameter $\mathbf{w}$, the expectation $\mathbb{E}$ is taken from sampling the input $x$ from data distribution $p_{data}$, and its pseudo-label $\hat{y}$ from the model distribution $r(\cdot|x)$. As the model distribution is different from the data distribution, the pseudo-gradient is different from the gradient used in SGD. In practice, the one-sample Monte-Carlo (1mc) approach (Martens & Grosse, 2015) is used to generate a pseudo-label $\hat{y}$ for each input $x$ from the model output distribution $r(\cdot|x)$, and then the pseudo-gradient is calculated via back-propagation (BP) w.r.t. the loss between model outputs and pseudo-labels. To avoid any confusion, the true FIM is usually denoted as $F_{1mc}$.

However, in many existing K-FAC implementations, it is a common practice to construct the empirical FIM (Osawa et al., 2019; 2020; Ueno et al., 2020; Pauloski et al., 2020; 2021) to avoid additional back-propagation computations with pseudo-labels for efficient training. The empirical FIM can be represented as

$$F_{emp} = \mathbb{E}_{(x,y) \sim p_{data}}[\mathtt{vec}(\nabla \ell)\mathtt{vec}(\nabla \ell)^T], \tag{18}$$

where $\nabla \ell = \nabla \log r(y|x)$ is the the log-likelihood gradient of a loss between the input $x$ and its label $y$ (sampled from the data distribution). As this gradient is exactly the same one in SGD, $F_{emp}$ can be efficiently constructed during the same feed-forward and back-propagation process of SGD. In this work, we focus on the analysis of the empirical FIM as it is more efficient than the true FIM while having the similar convergence performance with the true FIM in practise (Osawa et al., 2019; 2020; Ueno et al., 2020; Pauloski et al., 2020; 2021). Thus, throughout the following context, $F$ refers to $F_{emp}$ if not particularly specified.

**K-FAC.** Due to the large number of parameters in DNNs, it is impractical to store and invert the whole $F$, which takes $O(n^2)$ memory. Many approximation methods have been proposed to reduce the quadratic memory cost, such as Matrix-free methods (Martens et al., 2010; Frantar et al.,

2021), and Kronecker factorization (K-FAC) methods (Martens & Grosse, 2015; Grosse & Martens, 2016; George et al., 2018). In particular, K-FAC methods provide efficient and effective approximations of $F$, and have been successfully applied in large-scale distributed DNNs training for faster convergence (Osawa et al., 2019; Ueno et al., 2020; Pauloski et al., 2020; 2021).

Specifically, given an $L$-layer DNN, K-FAC approximates the full $F$ with a block-wise diagonal matrix as follows:

$$F \approx \operatorname{diag}(F_1, \cdots, F_L), \quad \text{where } F_l = \mathbb{E}[\operatorname{vec}(\nabla \ell_l)\operatorname{vec}(\nabla \ell_l)^T], \tag{19}$$

where $F_l$ is the FIM block constructed from the gradient information of layer $l$. For each $F_l$, K-FAC approximates it as the Kronecker product of two smaller matrices. Take a linear layer as an example, where $\mathbf{a}_l = \phi(W_l \mathbf{a}_{l-1})$, $F_l$ can be approximated by

$$F_l = \mathbb{E}[\operatorname{vec}(\nabla \ell_l)\operatorname{vec}(\nabla \ell_l)^T], \tag{20}$$

$$= \mathbb{E}[(\mathbf{a}_{l-1} \otimes \mathbf{g}_l)(\mathbf{a}_{l-1} \otimes \mathbf{g}_l)^T] = \mathbb{E}[(\mathbf{a}_{l-1}\mathbf{a}_{l-1}^T) \otimes (\mathbf{g}_l\mathbf{g}_l^T)], \tag{21}$$

$$\approx \mathbb{E}[\mathbf{a}_{l-1}\mathbf{a}_{l-1}^T] \otimes \mathbb{E}[\mathbf{g}_l\mathbf{g}_l^T] \triangleq A_{l-1} \otimes G_l. \tag{22}$$

The second-line derivation is based on the chain rule of $\mathbb{E}[\nabla \ell_l] = \mathbb{E}[\mathbf{g}_l\mathbf{a}_{l-1}^T]$ and the properties of Kronecker product $\otimes$, where $\nabla \ell_l$ is the gradient matrix w.r.t. the parameter matrix $W_l \in \mathbb{R}^{d_l \times d_{l-1}}$, $\mathbf{a}_{l-1} \in \mathbb{R}^{d_{l-1}}$ is the input vector of layer $l$ (i.e., activation of layer $l-1$), $\mathbf{g}_l \in \mathbb{R}^{d_l}$ is the gradient vector w.r.t. the pre-activation output of layer $l$, and $\phi$ is an element-wise non-linear activation function. The third-line approximation is based on the assumption that $\mathbf{a}_{l-1}$ and $\mathbf{g}_l$ are fairly independent (Martens & Grosse, 2015), which gives $\mathbb{E}[(\mathbf{a}_{l-1}\mathbf{a}_{l-1}^T) \otimes (\mathbf{g}_l\mathbf{g}_l^T)] \approx \mathbb{E}[\mathbf{a}_{l-1}\mathbf{a}_{l-1}^T] \otimes \mathbb{E}[\mathbf{g}_l\mathbf{g}_l^T]$. The $A_{l-1} = \mathbb{E}[\mathbf{a}_{l-1}\mathbf{a}_{l-1}^T]$ and $G_l = \mathbb{E}[\mathbf{g}_l\mathbf{g}_l^T]$ are called Kronecker factors (KFs). We notice that recent work points out K-FAC may not approximate FIM, but should be linked to gradient descent on neurons (Benzing, 2022). We are open to study this new perspective as well in the future work.

In summary, K-FAC approximates the full FIM diagonally via $F \approx \operatorname{diag}(F_1, \cdots, F_L)$, and then use the Kronecker product of two KFs to approximate each FIM block via $F_l \approx A_{l-1} \otimes G_l$. Based on the property of $(A \otimes G)\operatorname{vec}(V) = \operatorname{vec}(GVA)$, the preconditioned gradient is computed layer-wisely:

$$F_l^{-1}\operatorname{vec}(\nabla \ell_l) = (A_{l-1}^{-1} \otimes G_l^{-1})\operatorname{vec}(\nabla \ell_l) = \operatorname{vec}(G_l^{-1}\nabla \ell_l A_{l-1}^{-1}). \tag{23}$$

Thus, preconditioning the gradient in K-FAC is simply to left-multiply the gradient with $G_l^{-1}$ and right-multiply it with $A_{l-1}^{-1}$, where both KFs are symmetric matrices.

The K-FAC approximation reduces the memory complexity from a quadratic cost of storing FIM to a linear cost of storing KFs. Specifically, the total model parameter number is $n = \sum_{l=1}^{L}(d_{l-1} \times d_l)$, while the KF parameter number is $\sum_{l=1}^{L}(d_{l-1}^2 + d_l^2) = O(kn)$, in which $k \geq 2$. In addition, the K-FAC approximation is by no means limited to linear layers, and it has been successfully applied into other types of DNNs, such as CNNs, RNNs, and neural ODEs (Grosse & Martens, 2016; Martens et al., 2018; Liu et al., 2021).

**Damping technique.** To stabilize the second-order optimization, the damping technique (Moré, 1978) is commonly applied into the FIM (i.e., $F_l + \gamma I$ with a damping value $\gamma > 0$, before preconditioning. As inverting the large damped FIM directly is very expensive, K-FAC exploits an alternative approximate approach to invert two smaller damped KFs, i.e.,

$$(F_l + \gamma I)^{-1} = (A_{l-1} \otimes G_l + \gamma I)^{-1} \approx (A_{l-1} + \pi_l\sqrt{\gamma}I)^{-1} \otimes (G_l + \frac{\sqrt{\gamma}}{\pi_l}I)^{-1}, \tag{24}$$

where $\pi_l = \sqrt{T(A_{l-1})}/\sqrt{T(G_l)}$. $T(\cdot)$ is the trace of a square matrix divided by its dimension, which is derived to minimize the approximation error of inverting damped KFs (Martens & Grosse, 2015). To avoid that $\pi_l$ becomes infinitely large when $G_l$ is nearly zero, we restrict $\pi_l$ no larger than $10^9$ in practise. With the damping technique, the K-FAC update formula becomes:

$$W_l^{(t+1)} = W_l^{(t)} - \alpha^{(t)}(G_l + \frac{\sqrt{\gamma}}{\pi_l}I)^{-1}\nabla \ell_l(\mathbf{w}^{(t)}, \mathcal{B}^{(t)})(A_{l-1} + \pi_l\sqrt{\gamma}I)^{-1}, \tag{K-FAC}$$

where $I$ is an identity matrix. Therefore, K-FAC only needs to construct and invert damped KFs for all layers ($l = 1, \cdots, L$). The damped KFs are positive definite and they are typically inverted using Cholesky decomposition (Krishnamoorthy & Menon, 2013).

### A.3 K-FAC VARIANTS

The K-FAC algorithm can potentially improve the training performance by using a much less number of iterations than the SGD counterpart to reach the target accuracy (Osawa et al., 2019; Ueno et al., 2020). However, the iterative training time of K-FAC is much longer than that of SGD, as it needs to construct and invert KFs, which limits the practical usability of K-FAC. Distributed training techniques can also help alleviate the heavy inverse computations with multiple workers (Osawa et al., 2019; 2020; Pauloski et al., 2020; 2021). As distributed K-FAC introduces new bottleneck of communicating KFs, recent works (Shi et al., 2021; Zhang et al., 2022; 2023) were proposed to optimize its communication cost to reduce training time. Besides, the eigen-decomposition methods (George et al., 2018; Pauloski et al., 2020; 2021) target at providing better FIM approximations, for more stable optimization performance, by eigen-decomposing the KFs. For example, the *exact* damped preconditioner is given by eigen-decomposing KFs:

$$(F_l + \gamma I)^{-1} = (A_{l-1} \otimes G_l + \gamma I)^{-1} = (Q_A \otimes Q_G)(D_A \otimes D_G + \gamma I)^{-1}(Q_A^T \otimes Q_G^T), \quad (25)$$

where $A_{l-1} = Q_A D_A Q_A^T$ and $G_l = Q_G D_G Q_G^T$ are the orthogonal eigen-decompositions of two KFs. The eigen-decomposition provides the exact preconditioner rather than approximate preconditioner with matrix inversion (see Eq. 24). However, eigen-decomposition operations introduce much larger per-iteration computation overheads than matrix inversion operations (Zhang et al., 2022).

In addition, to alleviate the computational bottleneck, SKFAC (Tang et al., 2021) and SENG (Yang et al., 2022) applied the Woodbury formula to invert the Kronecker factors in a smaller dimension of mini-batch size, and KPSVD (Koroko et al., 2022) considered low-rank approximation of the Kronecker factors via expensive singular value decomposition. Unlike Eva, these methods attempt to approximate the low-rank FIM, and they are still compute-inefficient as they rely on either matrix inversion or decomposition operations.

**Stale FIM.** Existing K-FAC algorithms suffer from the system inefficiency due to the expensive overheads of computing and/or communicating KFs of all layers iteratively. As a compromise, they usually utilize the stale FIM information to increase the update interval of KFs to alleviate its inefficiency. However, skipping K-FAC approximations by using the stale statistics could bring potential negative effects on the convergence performance (Ba et al., 2017; Pauloski et al., 2020; Chen et al., 2021), and increase the complexity of tuning hyper-parameters (Ma et al., 2020).

## B    APPENDIX: DETAILED DERIVATION OF EVA

We provide the derivations of Eva from the trust-region optimization perspective, and supplement its algorithm and implementation details.

### B.1    TRUST-REGION OPTIMIZATION

The trust-region optimization provides a useful perspective toward understanding many second-order methods. Formally, we have

$$\mathbf{w}^{(t+1)} = \min_{\mathbf{w}} \ell(\mathbf{w}), \quad (26)$$

$$\text{s.t., } \rho(\mathbf{w}, \mathbf{w}^{(t)}) \leq \lambda, \quad (27)$$

where $\mathbf{w}^{(t)}$ is the current model parameter, $\ell$ is the loss function, and $\rho$ is the proximal function to measure the distance between two parameters. Therefore, the objective of trust-region optimization is to find the next model parameter $\mathbf{w}^{(t+1)}$ that minimizes the loss function while being close to the current parameter (i.e., in the trust region). It can be well solved by converting it into a proximal optimization problem (Bae et al., 2022) as follows:

$$\mathbf{w}^{(t+1)} = \min_{\mathbf{w}} \ell(\mathbf{w}) + \frac{1}{\lambda} \rho(\mathbf{w}, \mathbf{w}^{(t)}), \quad (28)$$

$$\approx \min_{\mathbf{w}} \ell(\mathbf{w}^{(t)}) + \text{vec}(\nabla \ell(\mathbf{w}^{(t)}))^T (\mathbf{w} - \mathbf{w}^{(t)}) + \frac{1}{2\lambda}(\mathbf{w} - \mathbf{w}^{(t)})^T P(\mathbf{w} - \mathbf{w}^{(t)}). \quad (29)$$

First, the proximal constraint of $\rho(\mathbf{w}, \mathbf{w}^{(t)}) \leq \lambda$ is added into the objective function as a penalty (smaller $\lambda$ means larger penalty). Second, we approximate the left-side loss term with first-order

expansion: $\ell(\mathbf{w}) \approx \ell(\mathbf{w}^{(t)}) + \text{vec}(\nabla\ell(\mathbf{w}^{(t)}))^T(\mathbf{w} - \mathbf{w}^{(t)})$, and approximate the right-side penalty term with second-order expansion: $\rho(\mathbf{w}, \mathbf{w}^{(t)}) \approx \frac{1}{2}(\mathbf{w} - \mathbf{w}^{(t)})^T P(\mathbf{w} - \mathbf{w}^{(t)}) \le \lambda$, where the metric $P$ depends on the choice of $\rho$.

As the approximated objective function is quadratic to $\mathbf{w}$, the optimal solution can be obtained as follows:

$$\mathbf{w}^{(t+1)} = \mathbf{w}^{(t)} - \lambda P^{-1}\text{vec}(\nabla\ell(\mathbf{w}^{(t)})). \tag{30}$$

This gives a generalized update formula of many second-order methods. For example, NGD uses the true FIM as the metric matrix, i.e., $P = F_{1mc}$, and the proximal constraint with the true FIM is in fact the second-order approximation of the KL-divergence between two model output distributions, i.e., $\text{KL}(p_\mathbf{w}||p_{\mathbf{w}^{(t)}}) \approx \frac{1}{2}(\mathbf{w} - \mathbf{w}^{(t)})^T F_{1mc}(\mathbf{w} - \mathbf{w}^{(t)})$.

## B.2 EVA

**Algorithm.** Following the same idea, Eva uses the damped outer-product of average gradient (OPAG) as the metric, that is

$$P = R + \gamma I = \mathbb{E}[\text{vec}(\nabla\ell)]\mathbb{E}[\text{vec}(\nabla\ell)]^T + \gamma I, \tag{31}$$

where $R = \mathbb{E}[\text{vec}(\nabla\ell)]\mathbb{E}[\text{vec}(\nabla\ell)]^T$ is the OPAG. Put it into the proximal function, we have

$$\rho(\mathbf{w}, \mathbf{w}^{(t)}) = \frac{1}{2}(\mathbf{w} - \mathbf{w}^{(t)})^T P(\mathbf{w} - \mathbf{w}^{(t)}) = \frac{1}{2}(\Delta\mathbf{w}^T \mathbb{E}[\text{vec}(\nabla\ell)])^2 + \frac{\gamma}{2}||\Delta\mathbf{w}||^2, \tag{32}$$

where $\Delta\mathbf{w} = \mathbf{w} - \mathbf{w}^{(t)}$ is the parameter update. Therefore, the proximal function consists of two terms: the first one constrains the parameter update along the gradient direction, and the second one limits the parameter update size. In other words, it is equivalent to combining the strip trust region and ball trust region, as shown in Fig. 2(c).

Like K-FAC, to reduce memory and computation costs, Eva approximates the OPAG with two smaller vectors at each DNN layer. Specifically, given a linear layer $\mathbf{a}_l = \phi(W_l \mathbf{a}_{l-1})$, the OPAG is approximated as follows:

$$R_l = \mathbb{E}[\text{vec}(\nabla\ell_l)]\mathbb{E}[\text{vec}(\nabla\ell_l)]^T = \mathbb{E}[\mathbf{a}_{l-1} \otimes \mathbf{g}_l]\mathbb{E}[\mathbf{a}_{l-1} \otimes \mathbf{g}_l]^T, \tag{33}$$

$$\approx (\mathbb{E}[\mathbf{a}_{l-1}] \otimes \mathbb{E}[\mathbf{g}_l])(\mathbb{E}[\mathbf{a}_{l-1}] \otimes \mathbb{E}[\mathbf{g}_l])^T \triangleq (\bar{\mathbf{a}}_{l-1} \otimes \bar{\mathbf{g}}_l)(\bar{\mathbf{a}}_{l-1} \otimes \bar{\mathbf{g}}_l)^T. \tag{34}$$

Similarly, the derivations are based on the chain rule of $\mathbb{E}[\nabla\ell_l] = \mathbb{E}[\mathbf{g}_l \mathbf{a}_{l-1}^T]$, and the assumption that $\mathbf{a}_{l-1}$ and $\mathbf{g}_l$ are *fairly* independent (Martens & Grosse, 2015). $\bar{\mathbf{a}}_{l-1} = \mathbb{E}[\mathbf{a}_{l-1}]$ and $\bar{\mathbf{g}}_l = \mathbb{E}[\mathbf{g}_l]$ are called Kronecker vectors (KVs). Since $R_l = \mathbf{v}_l\mathbf{v}_l^T$, where $\mathbf{v}_l = \bar{\mathbf{a}}_{l-1} \otimes \bar{\mathbf{g}}_l$, is a rank-one matrix, the inverse of the damped $R_l$ can be computed efficiently via the Sherman–Morrison formula (Sherman & Morrison, 1950). The Sherman–Morrison formula is

$$(A + \mathbf{u}_1\mathbf{u}_2^T) = A^{-1} - \frac{A^{-1}\mathbf{u}_1\mathbf{u}_2^T A^{-1}}{1 + \mathbf{u}_2^T A^{-1}\mathbf{u}_1}, \tag{35}$$

where $A$ is an invertible square matrix, $\mathbf{u}_1$ and $\mathbf{u}_2$ are column vectors. Taking $A = \gamma I$ and $\mathbf{u}_1, \mathbf{u}_2 = \mathbf{v}_l$ into the formula, we can invert the damped $R_l$ efficiently as follows:

$$(R_l + \gamma I)^{-1} = (\gamma I + \mathbf{v}_l\mathbf{v}_l^T)^{-1} = \frac{1}{\gamma}(I - \frac{\mathbf{v}_l\mathbf{v}_l^T}{\gamma + \mathbf{v}_l^T\mathbf{v}_l}) = \frac{1}{\gamma}(I - \frac{(\bar{\mathbf{a}}_{l-1}\bar{\mathbf{a}}_{l-1}^T) \otimes (\bar{\mathbf{g}}_l\bar{\mathbf{g}}_l^T)}{(\bar{\mathbf{a}}_{l-1}^T\bar{\mathbf{a}}_{l-1})(\bar{\mathbf{g}}_l^T\bar{\mathbf{g}}_l) + \gamma}). \tag{36}$$

Therefore, the preconditioned gradient with damped OPAG in the vector form is given by

$$(R_l + \gamma I)^{-1}\text{vec}(\nabla\ell_l) = \frac{1}{\gamma}\Big(\text{vec}(\nabla\ell_l) - \frac{\bar{\mathbf{g}}_l^T\nabla\ell_l\bar{\mathbf{a}}_{l-1}}{(\bar{\mathbf{a}}_{l-1}^T\bar{\mathbf{a}}_{l-1})(\bar{\mathbf{g}}_l^T\bar{\mathbf{g}}_l) + \gamma}\text{vec}(\bar{\mathbf{g}}_l\bar{\mathbf{a}}_{l-1}^T)\Big). \tag{37}$$

Thus, we can derive the update formula of Eva as:

$$W_l^{(t+1)} = W_l^{(t)} - \frac{\alpha^{(t)}}{\gamma}\Big(\nabla\ell_l - \frac{\bar{\mathbf{g}}_l^T\nabla\ell_l\bar{\mathbf{a}}_{l-1}}{(\bar{\mathbf{a}}_{l-1}^T\bar{\mathbf{a}}_{l-1})(\bar{\mathbf{g}}_l^T\bar{\mathbf{g}}_l) + \gamma}\bar{\mathbf{g}}_l\bar{\mathbf{a}}_{l-1}^T\Big). \tag{Eva}$$

**Relation to SGD.** Assume that $\mathbf{a}$ and $\mathbf{g}$ are *fully* independent, we have $\mathbb{E}[\texttt{vec}(\nabla\ell)] = \mathbb{E}[\mathbf{a} \otimes \mathbf{g}] = \mathbb{E}[\mathbf{a}] \otimes \mathbb{E}[\mathbf{g}]$. This means one can replace the Kronecker product of KVs by the gradient in Eva's update formula. That is, we can derive a simple update formula without using KVs as follows:

$$W_l^{(t+1)} = W_l^{(t)} - \frac{\alpha^{(t)}}{\gamma}(\nabla\ell_l - \frac{||\nabla\ell_l||^2}{(||\nabla\ell_l||^2 + \gamma}\nabla\ell_l) = W_l^{(t)} - \alpha^{(t)}\frac{\nabla\ell_l}{||\nabla\ell_l||^2 + \gamma}. \tag{38}$$

This shows that Eva (w/o KVs) uses only gradient norm to precondition the gradient in each DNN layer, behaving like a layer-wise adaptive SGD algorithm (You et al., 2018). However, the assumption that $\mathbf{a}$ and $\mathbf{g}$ are completely independent is unrealistic, because $\bar{\mathbf{g}}_l^T\bar{\mathbf{a}}_{l-1}$ is a rank-one matrix while the expectation of gradient is generally not (i.e., $\mathbb{E}[\nabla\ell_l] \neq \bar{\mathbf{g}}_l^T\bar{\mathbf{a}}_{l-1}$). Therefore, performing Eq. 38 directly will lose second-order information and affect the convergence performance compared to the Eva algorithm (see Table 5). Nevertheless, the Kronecker factorization of KVs are able to capture the structure information that helps optimization in deep learning training.

**KVs estimation for different types of layers.** The key component of Eva is to construct KVs used to precondition the gradient. Due to different structure of different types of layers, how to construct KVs is different. First of all, we consider how to estimate KVs for linear and convolutional layers.

*Linear layer without bias*: it is straightforward to estimate the KVs during feed-forward (FF) and back-propagation (BP) processes with a mini-batch of intermediate states by

$$\bar{\mathbf{a}}_{l-1} = \mathbb{E}[\mathbf{a}_{l-1}] = \frac{1}{|\mathcal{B}|}\sum_{i \in \mathcal{B}}\mathbf{a}_{l-1}^{(i)} \quad\text{and}\quad \bar{\mathbf{g}}_l = \mathbb{E}[\mathbf{g}_l] = \frac{1}{|\mathcal{B}|}\sum_{i \in \mathcal{B}}\mathbf{g}_l^{(i)}, \tag{39}$$

where $\mathbf{a}_{l-1}^{(i)}$ and $\mathbf{g}_l^{(i)}$ are input and pre-activation gradient vectors at layer $l$, sampled from the mini-batch of data $\mathcal{B}$. In practice, the mini-batch size used to estimate KVs can be a subset of the current batch. We find that a mini-batch size of 16 works well to estimate KVs.

*Linear layer with bias*: we can formulate it as matrix-vector multiplication:

$$\mathbf{s}_i = W_l\mathbf{a}_{l-1} + \mathbf{b}_l = (W_l \quad \mathbf{b}_l)\begin{pmatrix}\mathbf{a}_{l-1} \\ 1\end{pmatrix}, \tag{40}$$

where $W_l$ and $\mathbf{b}_l$ are weight and bias parameters, respectively. Therefore, to precondition the concatenated gradient of weight and bias, one should estimate $\bar{\mathbf{a}}_{l-1}$ over the input activation associated with one.

*Convolutional layer*: we need to convert the convolution operation into a linear transformation to estimate KVs. Assume that the input size is $(C_{in}, H_{in}, W_{in})$, and the output size is $(C_{out}, H_{out}, W_{out})$ in a conv layer, where $C$ denotes a number of channels, $H$ is a height in pixels, $W$ is a width in pixels, the convolution operation is defined as follows:

$$\text{out}[c_{out}] = \sum_{c_{in}=1}^{C_{in}}\text{weight}[c_{out}][c_{in}] \star \text{input}[c_{in}], \quad c_{out} = 1, \cdots, C_{out}, \tag{41}$$

where $\star$ is the 2D cross-correlation operator. There are totally $C_{in} \times C_{out}$ cross-correlation operations, and each $\star$ operation involves parameters with kernel size $K_H \times K_W$. By unfolding the input into a matrix $X$, one can perform convolution as a linear transformation (Chellapilla et al., 2006):

$$\text{out} = \text{reshape}(XW^T), \tag{42}$$

where $X$ is the unfolding input matrix of shape $H_{out}W_{out} \times C_{in}K_HK_W$ (each row contains all necessary values within the receptive fields), and $W^T$ is the weight matrix of shape $C_{in}K_HK_W \times C_{out}$ (each column contains all in-channel kernel weights). To construct KVs used to precondition the gradient of weight matrix, we are interested in the output pixel-wise linear transformation:

$$\mathbf{y}_i = W\mathbf{x}_i, \quad i = 1, \cdots, H_{out}W_{out}, \tag{43}$$

where $\mathbf{x}_i$ and $\mathbf{y}_i$ are the $i$-th row of the input matrix $X$ and output matrix $XW^T$. In other words, we can treat the pixel-wise out-channel computations as linear transformations on $H_{out}W_{out}$ vectors. Therefore, the KVs for a convolutional layer can be efficiently constructed as follows:

$$\bar{\mathbf{a}}_{\text{conv}} = \frac{1}{|\mathcal{B}|}\sum_{i \in \mathcal{B}}\text{mean}(X^{(i)}, 0) \quad\text{and}\quad \bar{\mathbf{g}}_{\text{conv}} = \frac{1}{|\mathcal{B}|}\sum_{i \in \mathcal{B}}\text{mean}(\nabla_{out}^{(i)}, [1, 2]), \tag{44}$$

where $X^{(i)}$ is the unfolding input sampled from $\mathcal{B}$, and $\nabla_{out}^{(i)}$ is the corresponding pre-activation output gradient tensor. The $\bar{\mathbf{a}}_{\text{conv}} \in \mathbb{R}^{C_{in} K_H K_W}$ and $\bar{\mathbf{g}}_{\text{conv}} \in \mathbb{R}^{C_{out}}$ take the average estimations over all rows of all inputs, and all pixels of all outputs, respectively. Like linear layers, the calculated KVs are used to precondition the gradient of the weight matrix. If bias is involved, one can simply add one element into $\bar{\mathbf{a}}_{\text{conv}}$ as discussed before.

**Gradient clipping.** After the gradient is preconditioned with KVs, the size of preconditioned gradient is typically an order of magnitude larger than the size of the original gradients (in terms of L2-norm), which could cause divergence. To ensure the model update is inside the trust region, different gradient clipping strategies can be applied to prevent exploding the preconditioned gradient. As suggested in (Pauloski et al., 2020), one can clip the preconditioned gradient when the KL size is higher than a threshold, which is measured by the damped OPAG (Martens & Grosse, 2015) as $\text{vec}(\mathcal{G}_l)^T (R + \gamma I)\text{vec}(\mathcal{G}_l) = \text{vec}(\mathcal{G}_l)^T \text{vec}(\nabla \ell_l)$, where $\mathcal{G}_l$ is the preconditioned gradient at layer $l$. Thus, one can scale the preconditioned gradients by a factor of

$$\nu_{KL} = \min\left(1, \sqrt{\frac{\kappa}{\alpha^2 \sum_{l=1}^{L} \text{vec}(\mathcal{G}_l)^T \text{vec}(\nabla \ell_l)}}\right), \tag{45}$$

where $\kappa > 0$ is the threshold for KL clipping. As KL clip is coupled with learning rate schedule algorithms, we find it works well for multi-step learning schedule following the practise of (Pauloski et al., 2020).

In addition, to avoid any hyper-parameter for clipping (i.e., $\kappa$), we propose gradient rescaling to keep the preconditioned gradient in the same size of the original gradient, that is, multiplying preconditioned gradient by a factor of

$$\nu_{GR} = \sqrt{\frac{\sum_{l=1}^{L} ||\nabla \ell_l||^2}{\sum_{l=1}^{L} ||\mathcal{G}_l||^2}}. \tag{46}$$

With gradient rescaling, preconditioning will not change the size of the gradient. We apply it in our finetuning experiments (see Table 6), and the results show that Eva can generalize on par with SGD under the same cosine learning rate schedule. As gradient clipping is of importance for preconditioned gradient descent algorithms such as K-FAC and Eva, we leave designing effective gradient clipping or scaling strategies as our future work.

**Implementation.** We implement Eva atop PyTorch. Following (Pauloski et al., 2020; 2021), we build a preconditioner to estimate KVs and precondition the gradients before performing the standard SGD optimizer. Our preconditioner supports Linear and Conv2D layers, and parameters at other layers (e.g., BatchNorm2d) will be updated by SGD without preconditioning. To construct KVs at supported layers, we register forward pre-hooks and backward-hooks to capture the activations and pre-activation gradients during the feed-forward and back-propagation computations, respectively. These intermediate values are used to estimate new KVs and update the running average states. Then we use stored KVs to precondition the gradients layer-wisely, and perform gradient clipping on preconditioned gradients, before they are used to update the model parameters in SGD.

To support data parallelism, we implement our Eva preconditioner to communicate KVs at each worker via *all-reduce* primitives. The communication of KVs is very efficient as the data volume of KVs is sublinear to the number of gradients, and small KVs can be merged to be communicated together via the tensor fusion technique supported by the distributed training framework Horovod (Sergeev & Del Balso, 2018). The aggregated KVs are then used to precondition the aggregated gradients on all workers. Unlike distributed K-FAC (Osawa et al., 2019; Shi et al., 2021), distributed Eva does not need to assign matrix-inversion tasks at different layers into different workers, and it also does not need to use stale FIM to skip the precondition of many iterations (e.g., update KFs every 50 iterations). Instead, it is memory- and time-efficient to construct KVs and precondition the gradients on all workers during the whole training process.

## C   Appendix: Supplement of Experiments

### C.1   Experimental settings

**Testbed.** We conduct our experiment on a 32-GPU cluster. It consists of 8 nodes connected 10Gb/s Ethernet, and each node has 4 Nvidia RTX2080Ti GPUs connected by two Intel(R) Xeon(R) Gold 6230 CPUs, 512GB memory, and PCIe3.0x16. We use some common software including PyTorch-1.10.0, Horovod-0.21.0, CUDA-10.2, cuDNN-7.6, and NCCL-2.6.4.

**Datasets and Models.**   We conduct our experiments on three commonly used datasets: Cifar-10 (Krizhevsky, 2009), Cifar-100 (Krizhevsky, 2009), and ImageNet (Deng et al., 2009). The Cifar-10/100 has 50,000 training images and 10,000 validation images. The ImageNet has ~1.3M training images and 50,000 validation images. On Cifar-10 and Cifar-100 datasets, we choose three representative models: VGG-19 (Simonyan & Zisserman, 2015), ResNet-110 (He et al., 2016), and WRN-28-10 (Zagoruyko & Komodakis, 2016). On the ImageNet dataset, we select ResNet-50 (He et al., 2016), Inception-v4 (Szegedy et al., 2017), and ViT-B/16 (Dosovitskiy et al., 2021) models.

**Baselines.**   We compare our Eva to the first-order baseline SGD, and second-order baseline K-FAC (Martens & Grosse, 2015). Besides SGD and K-FAC, we select two first-order adaptive gradient algorithms Adagrad (Duchi et al., 2010) and AdamW (Loshchilov & Hutter, 2019), and two other second-order algorithms M-FAC (Frantar et al., 2021) and Shampoo (Anil et al., 2021). We run each algorithm for 3 runs to compute the average metric (e.g., top-1 validation accuracy).

**Hyper-parameters**. We provide hyper-parameter configurations for reproducibility as below.

- In training VGG-19, ResNet-110, WRN-28-10 on Cifar-10 and Cifar-100 with SGD, K-FAC, and Eva, following (Pauloski et al., 2020), we set the mini-batch size to 512, learning rate to 0.4, and weight decay to 5e-4. We apply the multi-step learning rate schedule (a linear warmup at the first 5 epochs and learning rate decays by a factor of 10 at 35%, 75%, and 90% epochs). For K-FAC and Eva, we set damping to 0.03, running average to 0.95, and KL-clip to 0.001. The second-order update interval of K-FAC is 10.

- In training VGG-19, ResNet-110, WRN-28-10 on Cifar-10 with 4 more algorithms (Adagrad, AdamW, Shampoo, and M-FAC), we set the mini-batch size to 512, and tune the learning rate (from 10e-4 to 4) and choose the best one for each algorithm. Specifically, we set the learning rate as 0.4, 0.06, 0.04, and 0.004 for Shampoo, M-FAC, Adagrad, and AdamW, respectively. For AdamW, we set weight decay to 0.05 (5e-4 for others). For M-FAC, we set the number of gradient copies to 32. For Shampoo, we set the second-order update interval to 10, and apply the same multi-step learning rate schedule. We use cosine learning rate schedule (Loshchilov & Hutter, 2017) in other 3 algorithms (Adagrad, AdamW, and M-FAC) for better results.

- In time and memory efficiency comparison, we run Shampoo, K-FAC, and Eva on an Nvidia RTX2080Ti GPU. The second-order update interval of Shampoo and K-FAC is 10 on Cifar-10 (and 50 on ImageNet). On Cifar-10, we set the batch size as 512 for training VGG-19 and ResNet-110, and 256 for WRN-28-10. On ImageNet, we set the batch size as 64, 32, 16 for training ResNet-50, Inception-v4, and ViT-B/16, respectively. We measure the average time over 250 iterations, and obtain the GPU memory consumption by calling the "nvidia-smi" command.

- In the end-to-end performance comparison, we train VGG-19, ResNet-110, WRN-28-10 on Cifar-10 with one GPU. We set learning rate to 0.4, batch size to 512 for VGG-19, ResNet-110, and 256 for WRN-28-10. We run SGD for 200 epochs, and run Shampoo, K-FAC, and Eva for 100 epochs. We use the same multi-step learning rate schedule as before. For Shampoo and K-FAC, the update interval is 10. For K-FAC and Eva, the damping is 0.03 and the running average is 0.95. The target validation accuracy is 93% for training VGG-19 and ResNet-110, and 96.5% for training WRN-28-10.

- In end-to-end performance comparison, we train ResNet-50 on ImageNet-1k with 32 GPUs. To maximize the GPU memory, we set the per-GPU batch size to 64 for Shampoo and K-FAC, and to 96 for SGD and Eva. We set the learning rate to $0.05 \times 32 = 1.6$. Following (Pauloski et al., 2020), we train K-FAC and Eva for 55 epochs with learning rate decays at 25, 35, 40, 45, 50 epochs, and we set the damping to 0.001 and the running aver-

age to $0.95$. We train Shampoo for 60 epochs with tuned decays at 30, 45, 55 epochs, and SGD for 100 epochs, with tuned decays at 30, 60, 90 epochs. For Shampoo and K-FAC, the update interval is 50. The target validation accuracy is $75.9\%$ (MLPerf).

## C.2 ADDITIONAL EXPERIMENTAL RESULTS

**Autoencoder.** Following (Martens & Grosse, 2015; Goldfarb et al., 2020; Ren & Goldfarb, 2021), we compare the optimization performance of each algorithm by training an 8-layer autoencoder (with hidden dimensions of $[1000, 500, 250, 30, 250, 500, 1000]$) on three datasets: MNIST, FACES, and CURVES. We set batch size to 1000, and run each algorithm for 100 epochs with a linear decay learning rate schedule. We tune the learning rate in the range of $[0.001, 0.5]$ for different cases. The results are given in Fig. 5. The Fig. 5 shows that second-order methods K-FAC and Eva optimize much faster than SGD in this task, and our Eva can optimize the autoencoder model at the same convergence speed as K-FAC. In addition, one can see that Eva performs closely or better than another second-order method Shampoo, and Shampoo is faster than Adagrad except on CURVES. Note that Shampoo is a full-matrix version of Adagrad. We also compare the generalization performance of each algorithm in this task, and the results are very close to those of optimization comparison, that is, Eva performs similarly to K-FAC and outperforms other first-order counterparts on three datasets.

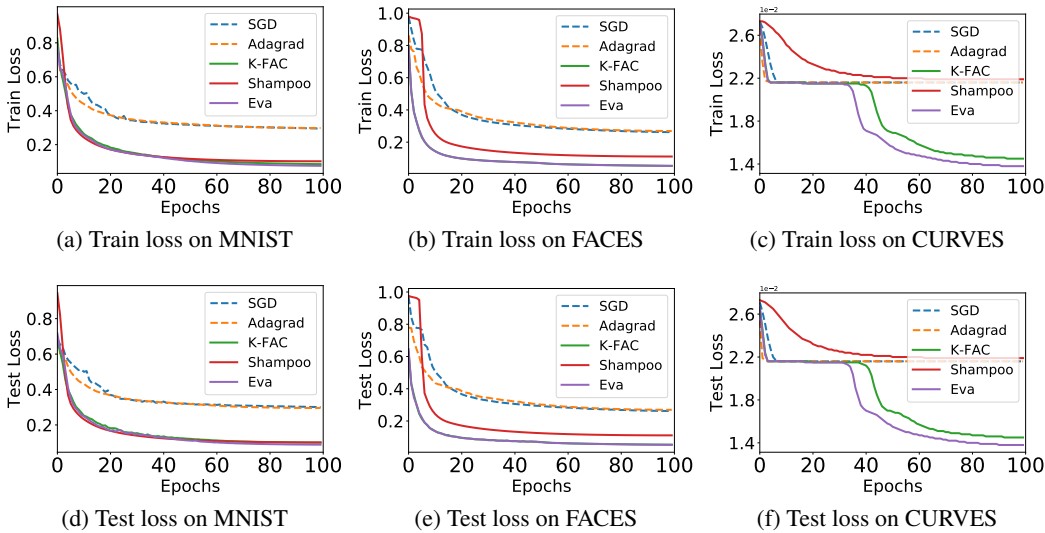

Figure 5: Optimizing an autoencoder on MNIST/FACES/CURVES with different algorithms.

**Training dynamics study.** To understand the training dynamics of Eva, we provide a case study by setting Eva's learning rate as $\alpha_{Eva} = \gamma \times \alpha_{SGD}$, and then update formula of Eva is given by

$$W_l^{(t+1)} = W_l^{(t)} - \alpha_{SGD}^{(t)} \left( \nabla \ell_l - \beta_l \cdot \bar{\mathbf{g}}_l \bar{\mathbf{a}}_{l-1}^T \right), \tag{47}$$

$$\text{where} \quad \beta_l = \frac{\bar{\mathbf{g}}_l^T \nabla \ell_l \bar{\mathbf{a}}_{l-1}}{(\bar{\mathbf{a}}_{l-1}^T \bar{\mathbf{a}}_{l-1})(\bar{\mathbf{g}}_l^T \bar{\mathbf{g}}_l) + \gamma}. \tag{48}$$

In this case, $\beta_l$ has played an important role in Eva, making its training dynamics different from SGD (which can be viewed as $\beta_l = 0$). Here we optimize the autoencoder following the original setting, except that we set $\alpha_{SGD} = 0.6$, $\alpha_{Eva} = 0.018$, $\gamma = 0.03$, and disable the KL-clip for Eva.

The optimization performance comparison is given in Fig. 6(a), showing that Eva can still outperform SGD under the condition that $\alpha_{Eva} = \gamma \times \alpha_{SGD}$. This implies that $\beta_l \cdot \bar{\mathbf{g}}_l \bar{\mathbf{a}}_{l-1}^T$ can help precondition the gradient information. Therefore, we also plot the values of $\beta_l$ in each layer during the training process in Fig. 6(b), showing that $\beta_l$ are very small ($< 10^{-3}$) but not zeros, and $\beta_l$ are gradually decreasing. This implies that Eva preconditions more in the early stage, but acts more like SGD in the late stage. Besides, $\beta$ values are adaptive to different layers, e.g., layer 7 seems the most ill-conditioned as it requires the largest $\beta$ values for preconditioning.

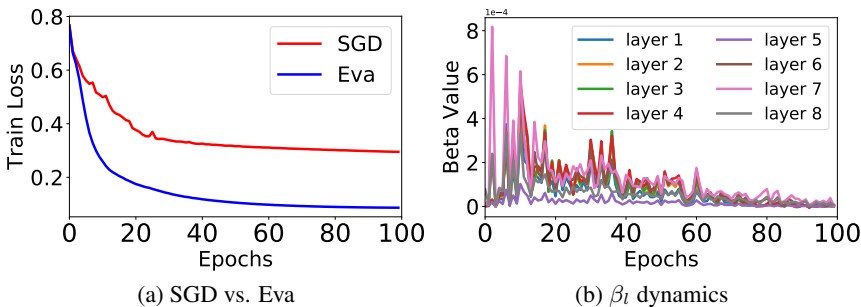

(a) SGD vs. Eva        (b) $\beta_l$ dynamics

Figure 6: Training dynamics study of Eva in optimizing an autoencoder on MNIST.

**Convergence results with 50 epochs.** To verify that second-order algorithms K-FAC and Eva can learn faster than first-order algorithms such as SGD, especially in the early epochs (Frantar et al., 2021), we report the training loss and validation error curves (log-scale), in Fig. 7, for training ResNet-110 on Cifar-10 and VGG-19 on Cifar-100 with 50 epochs. The results show that K-FAC and Eva converge very closely, and they indeed outperform SGD on both optimization and generalization abilities. As studied in Table 2, K-FAC and Eva can outperform SGD under different epoch budgets.

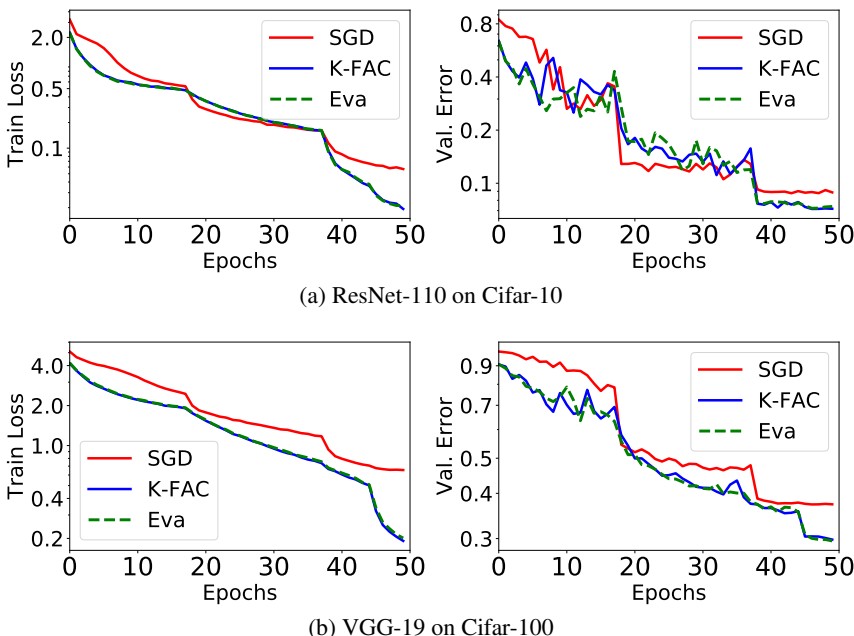

(a) ResNet-110 on Cifar-10

(b) VGG-19 on Cifar-100

Figure 7: Convergence performance (log-scale) comparison between Eva and SGD/K-FAC algorithms for training ResNet-110 on Cifar-10, and VGG-19 on Cifar-100 in compressed 50 epochs.

**Pairwise t-test comparison.** In addition to Table 2, we perform pairwise t-test with Bonfereonni correction for the accuracy metric among SGD, K-FAC, and Eva algorithms. The results show that Eva performs closely to K-FAC in all cases (i.e., they have no significant difference with $p > 0.05$). K-FAC and Eva significantly outperform SGD with $p < 0.05$ in many cases (12 out of 18 cases), including all models with 50 epochs, VGG-19 and ResNet-110 with 100 epochs, and VGG-19 with 200 epochs on two datasets. K-FAC and Eva are slightly better than SGD in other cases while SGD is trained with sufficient epochs and/or extra tricks.

**Finetuning pretrained models.** Pretraining an model on a large dataset and then finetuning on the downstream task (i.e., transfer learning) is a common practice to produce better model accuracy on small datasets. To further verify the effectiveness of Eva in this task, we choose two repre-

sentative pretrained models, EfficientNet-b0 (Tan & Le, 2019) (pretrained on ImageNet-1k) and ViT-B/16 (Dosovitskiy et al., 2021) (pretrained on ImageNet-21k). We load their weights from publicly available checkpoints, except the last classification layer, which is randomly initialized. As suggested in (Tan & Le, 2019; Dosovitskiy et al., 2021), we scale down initial learning rate to 0.04 and 0.004 for EfficientNet-b0 and ViT-B/16, set the batch size to 96, and use the cosine learning rate schedule (Loshchilov & Hutter, 2017). The weight decay is set to 5e-5 for EfficientNet-b0, and 0 for ViT-B/16, and the input images are resized to 224-pixel. For K-FAC and Eva, we set damping value to $0.03$ and running average to $0.95$. Gradient rescaling is also applied. We finetuning the models with 20 epochs. The results are given in Table 6, showing that second-order optimization methods such as K-FAC and Eva can generalize as well as SGD on finetuning pretrained models, even though they were pretrained with first-order algorithms (SGD for EfficientNet-b0, and AdamW for ViT-B/16). Specifically, Eva can achieve very competitive performance, i.e., 98.88% and 92.88% top-1 accuracy on Cifar-10 and Cifar-100, respectively, when finetuning ViT-B/16.

Table 6: Validation accuracy (%) comparison between Eva and SGD/K-FAC algorithms for finetuning pretrained models with 20 epochs. EffNet stands for EfficientNet.

| Model | Cifar-10 | | | Cifar-100 | | |
|---|---|---|---|---|---|---|
| | SGD | K-FAC | Eva | SGD | K-FAC | Eva |
| EffNet-b0 | $97.39_{\pm 0.1}$ | $97.37_{\pm 0.0}$ | $\mathbf{97.43}_{\pm 0.1}$ | $\mathbf{85.41}_{\pm 0.0}$ | $85.32_{\pm 0.2}$ | $85.38_{\pm 0.0}$ |
| ViT-B/16 | $98.87_{\pm 0.0}$ | $98.87_{\pm 0.0}$ | $\mathbf{98.88}_{\pm 0.0}$ | $92.79_{\pm 0.1}$ | $92.68_{\pm 0.2}$ | $\mathbf{92.88}_{\pm 0.0}$ |

**The effects of update interval.** As second-order algorithms such as K-FAC have expensive second-order update costs, it is necessary to increase the second-order update interval to achieve faster convergence (Martens & Grosse, 2015). Here we compare the end-to-end performance of K-FAC with different update intervals.

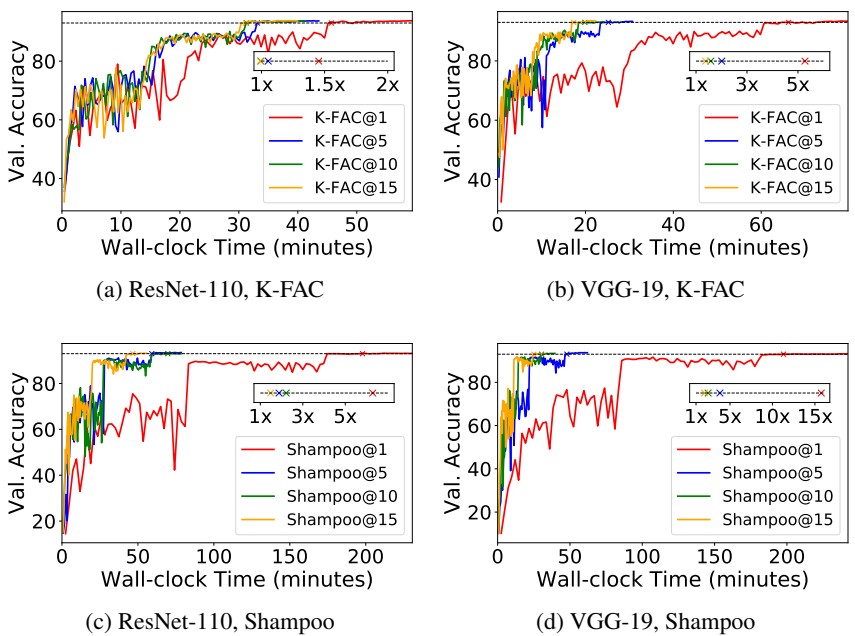

(a) ResNet-110, K-FAC

(b) VGG-19, K-FAC

(c) ResNet-110, Shampoo

(d) VGG-19, Shampoo

Figure 8: Wall-clock time comparison of K-FAC and Shampoo with different update intervals on Cifar-10. The inset plot reports relative time-to-solution over Eva.

As shown in Fig. 8, K-FAC@1 takes much longer training time than Eva, i.e., $1.45\times$ and $5.27\times$ than Eva for ResNet-110 and VGG-19, respectively. Thus, it is often required to increase the second-order update interval to make K-FAC more affordable. For example, the training speed of K-FAC@10 (update KFs and their inverses every 10 iterations) is comparable to Eva for training ResNet-100, and it is $1.58\times$ slower than Eva for training VGG-19 to achieve the target accuracy.

Unlike K-FAC, our Eva can converge quickly using an update interval of second-order information of 1, i.e., updating second-order information iteratively, which avoids the efforts of tuning the interval and the danger of performance degradation with stale information.

We also include the effects of update intervals on Shampoo for training ResNet-110 and VGG-19 on Cifar-10, in Fig. 8(c) and (d), respectively. The results show that increasing the update intervals of Shampoo can accelerate the training process with little performance degradation, but keep increasing the update interval only brings marginal performance improvement.

**The effects of batch size on throughput.** As Eva is more memory-efficient than other second-order algorithms, it is of interest to investigate the effects of batch size on system throughput. For that purpose, we train the ResNet-50 model with SGD, K-FAC@50, Shampoo@50, and Eva on ImageNet with 32 GPUs. We set the per-GPU batch size to maximally utilize GPU memory, i.e., 64 for K-FAC@50 and Shampoo@50, and 96 for SGD and Eva (64 for SGD and Eva is also conducted for comparison). The system throughput is given in Table 7. It shows that Eva, K-FAC@50, and Shampoo@50 can achieve 90%, 86%, and 68% throughput over SGD, when they adopt the same batch size of 64. However, as we scale the batch size of SGD and Eva to 96 (as they are more memory efficient than K-FAC and Shampoo), Eva achieves a much closer throughput (92%) with SGD, while the performance gap between other second-order algorithms (K-FAC@50 and Shampoo@50) and SGD becomes larger.

Table 7: Throughput (samples per second) comparison between Eva and other algorithms

| Algorithm | SGD | Shampoo@50 | K-FAC@50 | Eva | SGD | Eva |
|---|---|---|---|---|---|---|
| Batch Size | 64 | 64 | 64 | 64 | 96 | 96 |
| Throughput | 6420.1 | 4366.7 | 5520.2 | 5801.7 | 7420.3 | 6857.1 |

In addition to Fig. 3(d), we have added Table 8 to indicate the required number of epochs to achieve the target accuracy of ResNet-50 on ImageNet as follows.

Table 8: Number of epochs to achieve the target accuracy of ResNet-50 on ImageNet.

| Algorithm | SGD | Shampoo | K-FAC | Eva |
|---|---|---|---|---|
| Number of Epochs | 91 | 55 | 46 | 46 |

**The effects of hyper-parameters on Cifar-100.** We conduct hyper-parameter study of Eva, including learning rate, batch size, damping, and running average, by training VGG-19 on Cifar-100. The results are given in Fig. 9, showing that Eva can consistently outperform SGD under different learning rate and batch size, and Eva performs closely to K-FAC under different learning rate, batch size, damping and running average settings. Besides, both SGD and Eva performs poorly with a large learning rate, but Eva performs much better than SGD with a large batch size. We leave the study on the effects of large-batch training with second-order optimization methods (Zhang et al., 2019) as our future work. As for damping and running average hyper-parameters, we found that damping is more sensitive than running average, so that one can tune the damping value to further boost the performance of Eva.

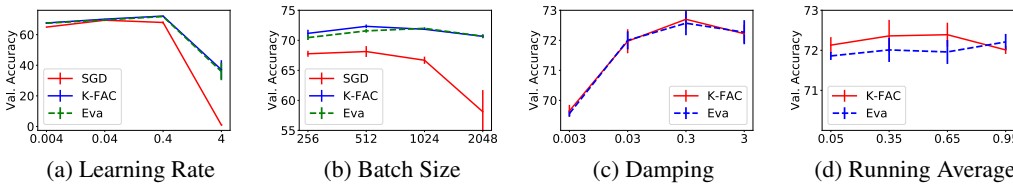

(a) Learning Rate · (b) Batch Size · (c) Damping · (d) Running Average

Figure 9: Hyper-parameter study of Eva by training VGG-19 on Cifar-100 with 100 epochs.

