# OpenReview forum: "Eva: Practical Second-order Optimization with Kronecker-vectorized Approximation"
_ICLR.cc/2023/Conference — ICLR 2023 poster_

### Official Review · Reviewer_1mYD · 2022-10-21

**Confidence:** 4
**Correctness:** 4
**Technical Novelty And Significance:** 3
**Empirical Novelty And Significance:** Not applicable
**Recommendation:** 5

**Clarity, Quality, Novelty And Reproducibility:**

The paper is very clearly written. While rank 1 approximations to large matrices are quite common, I am not aware of this approximation to K-FAC before.

**Strength And Weaknesses:**

Strengths: The paper is clearly written, and the proposed algorithm is very simple and easy to implement. The update rule is very fast to compute, and the algorithm uses very little memory, so is very fast. The experiments for CIFAR-10 are quite comprehensive.

Weaknesses: The approximations made in the algorithm are not motivated in the paper. Why is it acceptable to approximate E[a_{l-1}a_{l-1}^T] as E[a_{l-1}] ⊗ E[a_{l-1}]^T? K-FAC's approximations are justified by independence assumptions, but Eva's further approximations are not justified anywhere. There is no convergence analysis to show that this update is mathematically well grounded.

For Imagenet, please include a table showing the number of epochs needed to achieve target accuracy. Also for K-FAC and Shampoo, please include results of increasing the update interval further, since their authors say that larger intervals do not degrade accuracy very much. Also for the autoencoders, please include train error and test accuracy for the 3 common datasets --- MNIST, Faces and Curves.

It is strange that you claim that "K-FAC and Shampoo are not time-and-memory efficient,  which severely hinders their feasibility in practise (sic)" --- these optimizers are used to train some of the largest networks today, with billions of parameters.

K-FAC and Eva are both presented for fully connected and convolutional networks --- does Eva extend easily to other networks such as transformers, and to networks with features like batch norm or layer norm?

**Summary Of The Paper:**

The paper presents an optimizer in the K-FAC family. K-FAC approximates the Fisher information matrix as the Kronecker product of two smaller matrices, by assuming independence --- they approximate E((a_{l−1} ⊗ g_l)(a_{l-1} ⊗ g_l)^T] as E[a_{l-1}a_{l-1}^T] ⊗ E[g_l g_l^T ]. The proposed algorithm, Eva, goes one step further, by approximating E[a_{l-1}a_{l-1}^T] as E[a_{l-1}] ⊗ E[a_{l-1}]^T and E[g_l g_l^T] as
E[g_l]  ⊗ E[g_l]^T. The result is a rank one approximation to the Kronecker factors, and the authors use damping and the Sherman-Morrison-Woodbury formula to compute the preconditioner. Note that Eva approximates the empirical Fisher information matrix, as opposed to the true Fisher recommended by the K-FAC authors.

The algorithm uses significantly less memory than K-FAC, in fact uses less memory than Adam or Adagrad --- its memory consumption is similar to Adafactor or SM3. The paper has several comparisons on image datasets, showing that Eva is competitive with K-FAC and Shampoo in terms of accuracy, while achieving performance similar to SGD.

**Summary Of The Review:**

The paper presents an interesting and fast approximation to K-FAC. I would like to see some theoretical justification for the approximation, and also some experimental results as detailed above for a higher score.

---

> ### Author Response · Authors · 2022-11-07
> **Response to Reviewer 1mYD**
>
> **Question 1**: The approximations made in the algorithm are not motivated in the paper. Why is it acceptable to approximate $E[a_{l-1}a_{l-1}^T]$ as $E[a_{l-1}]E[a_{l-1}]^T$? K-FAC's approximations are justified by independence assumptions, but Eva's further approximations are not justified anywhere. There is no convergence analysis to show that this update is mathematically well grounded.
>
> **Response**: There could be some ambiguities in the manuscript. We would like to clarify that we do not try to approximate $E[a_{l-1}a_{l-1}^T]$ as $E[a_{l-1}] E[a_{l-1}]^T$ (which is not true according to Eq.(13)). Instead, we adopt the same assumptions from K-FAC, that is, $a_{l-1}$ and $g_{l}$ are fairly independent, as shown in Eq.(11). In another word, Eva does not try to approximate empirical FIM (F), but it approximates the second-order matrix (R) that satisfies $R \preceq  F$, implying Eva is a more aggressive trust-region algorithm than K-FAC as shown in Eq.(14).
>
> **Question 2**: For Imagenet, please include a table showing the number of epochs needed to achieve target accuracy.
>
> **Response**: Thanks for the suggestion. In addition to Figure 3(d),  we have added such a table (see Table 8 in the Appendix C.2) to indicate the required number of epochs to achieve the target accuracy of ResNet-50 on ImageNet as follows.
> | Algorithm        | SGD | Shampoo | K-FAC | Eva |
> |------------------|-----|---------|-------|-----|
> | Number of Epochs | 91  | 55      | 46    | 46  |
>
> **Question 3**: For K-FAC and Shampoo, please include results of increasing the update interval further, since their authors say that larger intervals do not degrade accuracy very much.
>
> **Response**: We have studied the effects of update intervals in Figure 8 in Appendix C.2, and the results show that, at least on Cifar-10, increasing the intervals does not degrade accuracy for K-FAC and Shampoo. We adopt the update intervals from previous work [2], e.g., 10 on Cifar-10, since further increasing the update interval only brings marginal performance improvement, as shown in Figure 8.
>
> **Question 4**:  For the autoencoders, please include train error and test accuracy for the 3 common datasets --- MNIST, Faces and Curves.
>
> **Response**: We have included train loss and test loss for the autoencoder on MNIST, Faces, and Curves datasets, given in Figure 5 in the Appendix C.2. The results show that Eva performs similarly to K-FAC (and better than K-FAC on Curves), and Eva can consistently outperform first-order counterparts in the optimization and generalization performance on different datasets.
>
> **Question 5**: It is strange that you claim that "K-FAC and Shampoo are not time-and-memory efficient, which severely hinders their feasibility in practise" --- these optimizers are used to train some of the largest networks today, with billions of parameters.
>
> **Response**: There is some ambiguity in this statement. The claim that K-FAC and Shampoo are not time-and-memory efficient is to consider their higher time and memory complexity when compared to first-order methods, which may make the end-to-end training performance of K-FAC and Shampoo be unable to enjoy their fast convergence properties. Practically, according to [1][2] and our experimental results (Figure 3), dedicated system optimizations and increasing the second-order update interval are required to make K-FAC and Shampoo be possibly better than SGD. According to the suggestion, we have changed the statement in the revised manuscript as “One limitation in K-FAC and Shampoo is that they typically require dedicated system optimizations and second-order update interval tuning to outperform the first-order counterpart.”
>
> **Question 6**: K-FAC and Eva are both presented for fully connected and convolutional networks --- does Eva extend easily to other networks such as transformers, and to networks with features like batch norm or layer norm?
>
> **Response**: Eva can be used to train Transformers. For instance, we have used Eva to finetune the pretrained ViT-B/16 model as shown in Table 6 in the Appendix C.2. To the networks with batch norm or layer norm, we follow the common practise of K-FAC [2] to just skip preconditioning batch norm or layer norm layers. As also suggested at the Limitation 1 by Reviewer j8t4, we will study the convergence performance of our Eva at different tasks in our future work.
>
> [1] Kazuki Osawa, Yohei Tsuji, Yuichiro Ueno, Akira Naruse, Rio Yokota, and Satoshi Matsuoka. Large-scale distributed second-order optimization using kronecker-factored approximate curvature for deep convolutional neural networks. CVPR, 2019.
>
> [2] J Gregory Pauloski, Zhao Zhang, Lei Huang, Weijia Xu, and Ian T Foster. Convolutional neural network training with distributed K-FAC. SC, 2020.

---

### Official Review · Reviewer_j8t4 · 2022-10-23

**Confidence:** 4
**Correctness:** 3
**Technical Novelty And Significance:** 2
**Empirical Novelty And Significance:** 3
**Recommendation:** 6

**Clarity, Quality, Novelty And Reproducibility:**

This paper is written clearly. The experimental details and code are available. Since there have already been many attempts at low-rank and rank-1 approximations of layer-wise Fisher for natural gradient, the technical novelty of Eva (a rank-1 approximation that averages a batch of activations/errors) is limited. However, there is an empirical novelty in that such a simple method can achieve the same convergence as K-FAC and other second-order optimization methods for various tasks, including ImageNet classification.

**Strength And Weaknesses:**

Strengthes
- The paper clearly describes the literature, the differences between the proposed and existing methods, and the experimental details, which are easy to follow.
- The proposed Eva method is straightforward (in a good sense). It boosts the practicality of second-order optimization in deep learning regarding computational and memory cost and ease of implementation and integration (as long as the fast convergence holds for other tasks). Thanks to the low computational cost, there is no need to use stale second-order information or to tune the second-order information update interval.
- Experiments are well designed to validate the effectiveness of second-order optimization methods.
    - (There are still missing points as I will describe in “Weaknesses”, though.)
    - Several numbers of epochs are considered. A longer training (200 epochs) gives the baseline first-order optimization methods a chance to achieve the same accuracy as the second-order methods. Yet, second-order optimization methods (K-FAC, Eva) show a faster convergence (It would be better if the same comparison (with 50, 100, and 200 epochs) is made for Shampoo and M-FAC, though.)
    - Hyperparameters specific to second-order optimization are explored. Since damping, second-order information update interval, and running average coefficient largely affect the convergence of mini-batch second-order optimization, a comparison with various values of them is valuable in verifying Eva’s robustness.

Weaknesses
- As the authors mention, there is no theoretical analysis in this study, and the effectiveness of Eva is supported only by the experimental results.
    - Eva has shown good experimental performance (as convergent as K-FAC). I appreciate the detailed experiments and analysis in this work. However, there is still no guarantee that good performance will hold for other tasks due to the algorithm's simplicity.
    - (Here, I am not requesting additional experiments but pointing out that the “no guarantee” is almost inevitable in deep learning, which involves many ‘unpredictable factors.’)
    - Therefore, I do not think comparing only the computation and memory costs of existing methods and Eva (as in Table1) is fair. The amount of information (e.g., number of matrix elements and matrix rank) that each method uses for preconditioning gradient should be included as a comparison item. Eva’s computational and memory cost is lower than K-FAC simply because it calculates/stores much less amount of information —- it approximates the Kronecker factor matrices with rank-1 matrices (outer products of the Kronecker vectors). I think readers would appreciate it if this work also shows the trade-off between the amount of information and the computational/memory cost (a ‘predictable factor’) to decide which algorithm to use.
- Comparison between K-FAC and Eva with different mini-batch sizes is missing.
    - (Here, I do request for additional experiments)
    - The sum of the outer product of average gradient (OPAG), aka mini-batch empirical FIM, is known to have less second-order information when the mini-batch size is larger (https://www.cs.toronto.edu/~rgrosse/courses/csc2541_2022/readings/L05_normalization.pdf, equation 4) unlike the (per-example) empirical FIM.
    - And again, since Eva approximates the Kronecker factor matrices with rank one, it should miss lots of information in large-batch settings (whether and how much the information is beneficial for training convergence and generalization is unpredictable, though).
    - Therefore, Eva, which uses OPAG or mini-batch empirical FIM, might underperforms K-FAC w/ empirical FIM in large-batch settings so I would appreciate more empirical comparison with different mini-batch sizes.
    - Regardless of the empirical results, it would be valuable to indicate the amount of information Eva captures/misses clearly.
- Comparison with several existing studies in the literature are missing.
    - SKFAC (https://openaccess.thecvf.com/content/CVPR2021/papers/Tang_SKFAC_Training_Neural_Networks_With_Faster_Kronecker-Factored_Approximate_Curvature_CVPR_2021_paper.pdf) and SENG (https://link.springer.com/article/10.1007/s10915-022-01911-x) also apply Sherman-Morrison-Woodbury (SMW) formula to invert the Kronecker factors. Both approaches approximate layer-wise Fisher matrix with low-rank matrix. So one can think of them as higher-rank counterpart of Eva.
    - KPSVD (https://hal.archives-ouvertes.fr/hal-03541459/document) also consider rank-one approximation of the Kronecker factor. Unlike Eva, it tries to find the optimal rank-one matrices that minimizes the Frobenius norm of the difference between the layer-wise Fisher matrix and the rank-one approximation. Compared to Eva, it might have more computational costs, but it sounds more theoretically-grounded. Again, whether this makes a change is unpredictable in practice, but showing the information-cost trade-off is useful.

**Summary Of The Paper:**

This study proposes Eva, a second-order optimization method for training deep neural networks that approximates the Kronecker factor of K-FAC using empirical FIM with a rank one matrix. Averaging batch activations or errors construct the rank one matrix. Experimental results on various computer vision tasks show that Eva achieves similar convergence (number of steps vs. accuracy) as K-FAC, Shampoo, and M-FAC. Since Eva's computational cost per step is considerably smaller than other second-order optimization methods and is close to SGD’s cost, Eva achieves relatively fast training times in wall-clock time.


**Summary Of The Review:**

Although there are no solid theoretical guarantees, a simple, low-cost method like Eva can achieve the same level of convergence as a high-cost method like K-FAC is a critical data point for future research on efficient training methods in deep learning. However, due to its simplicity, it is unpredictable that Eva is effective in a wide range of tasks. I believe that comparisons and discussions focusing on predictable aspects such as trade-offs between the amount of information (e.g., number of matrix elements, matrix rank to be used for gradient preconditioning) and computational/memory cost would enhance the value of this study.

---

> ### Author Response · Authors · 2022-11-07
> **Response to Reviewer j8t4**
>
> We appreciate the detailed and constructive comments from this reviewer.
>
> **Limitation 1**: Eva has shown good experimental performance (as convergent as K-FAC). I appreciate the detailed experiments and analysis in this work. However, there is still no guarantee that good performance will hold for other tasks due to the algorithm's simplicity. (Here, I am not requesting additional experiments but pointing out that the “no guarantee” is almost inevitable in deep learning, which involves many ‘unpredictable factors.’)
>
> **Response**: We certainly agree that a single optimizer could not guarantee to be the best for different types of tasks just like SGD is good for CNNs while Adam is good for Transformers. We are considering conducting more experiments on other tasks such as NLP using LSTMs and Transformers in our future work.
>
> **Limitation 2**: I do not think comparing only the computation and memory costs of existing methods and Eva (as in Table1) is fair. The amount of information (e.g., number of matrix elements and matrix rank) that each method uses for preconditioning gradient should be included as a comparison item.
>
> **Response**: This is a very valid comment. Based on the results that the convergence speed in terms of iterations is quite similar between Eva and K-FAC, Table 1 mainly shows the cost for memory and computation for a single update iteration. Thus, we only compare memory costs and computational complexity. However, as the memory cost is the same as the amount of information (number of matrix elements) that each method uses for preconditioning, it is true that reducing the memory cost can inevitably lose some information. Thus, it would make it more complete to include which second-order information is used for preconditioning as suggested (in Table 1). However, it is worth noticing that the amount of information (number of matrix elements) for preconditioning is not directly related to the convergence speed.
>
> **Limitation 3**: Comparison between K-FAC and Eva with different mini-batch sizes is missing.
>
> **Response**: We have added the comparison between K-FAC and Eva with different mini-batch sizes in Figure 4(b) (as well as Figure 9(b) in Appendix C.2), and the results show that Eva performs closely to K-FAC under different mini-batch sizes on different models and datasets.
>
> **Limitation 4**: Comparison with several existing studies in the literature are missing, such as SKFAC, SENG, and KPSVD.
>
> **Response**: Thanks for pointing out relevant work. We have added the comparison in the revised manuscript. Specifically, SKFAC and SENG applied the Woodbury formula to invert the Kronecker factors in a smaller dimension of batch size, and KPSVD considered low-rank approximation of the Kronecker factors via singular value decomposition. Unlike Eva, these methods attempt to approximate the low-rank FIM, and they are still computation-inefficient as they rely on either matrix inversion or decomposition operations.

---

> > ### Comment · Reviewer_j8t4 · 2022-11-15
> > **I appreciate the revision, but would like to keep the score.**
> >
> > Thank you for your reply.
> >
> > I appreciate additional experiments (comparison of K-FAC vs. Eva with varying mini-batch sizes) and additional text (literature, second-order information on each method). I also appreciate the additional sentence describing a limitation of K-FAC and Shampoo (they require dedicated system optimizations and second-order information update interval tuning). This point is critical from a practical point of view.
> >
> > As already mentioned, the experiments in this study are well designed, and it is a fascinating data point that Eva (layer-wise rank-2 approximation of the FIM) achieves fast training convergence in practice. However, I want to state again that it is hard to expect that the layer-wise FIM approximation with KVs is always valid (i.e., the training convergence is as fast as other second-order methods with more matrix ranks) for various neural networks, tasks, and training settings without any compelling theory/intuition (e.g., which elements of second-order information play a key role in training convergence and *generalization*, and why layer-wise rank-2 approximation can retain those elements). Additional training results are helpful but not critical to justify the validity of Eva's approximation.
> >
> > I believe that the authors have addressed some of my concerns and improved the paper's quality. However, I would like to keep the score for the above reason.

---

### Official Review · Reviewer_p2Vi · 2022-10-24

**Confidence:** 4
**Correctness:** 4
**Technical Novelty And Significance:** 3
**Empirical Novelty And Significance:** 3
**Recommendation:** 8

**Clarity, Quality, Novelty And Reproducibility:**

Clarity, quality and reproducibiltiy are all top notch, code is included and seems clean on a brief skim, as I noted, clarity is a particularl strength and quality of evaluation is high as well.

The only challenge I could make to this paper is that I see very little *theoretical* difference between this and K-FAC (it could be argued it's taking K-FACs approximation to a logical extreme) *but* the implementation details (Sherman-Morrison, rescaling, the exact manner of sampling) which follows the *practical* use of these methods compensates for this in my opinion.

**Strength And Weaknesses:**

Strengths:

- the explanation of related work, background etc. is well done, I think this paper can serve as a good entrypoint for readers
- the method is conceptually simple, but clever. I think it is non-obvious that the OPAG approximation would work as well as the K-FAC approximation a priori, and the exploitation of Sherman-Morrison and the rescaling to gradient norm in order to make the method drop-in are nice touches
- the experimental evaluation is top notch, using the related work and real world datasets and architectures
- the limitations section is an actual, honest self assessment

Weaknesses

- the authors state most weaknessses in the related work
- truly a nitpick, but statistical significance tests between results using bonferonni correction would have been a cherry on top

**Summary Of The Paper:**

The paper describes what I would call a (simple but clever) evolution of K-FAC style fisher information matrix approximation (FIM) by approximating the FIM with  kronecker products of activations and gradients (i.e. an outer product over vectors) where K-FAC uses a kronecker product of *matrices* derived from the activation and gradient. While this is an additional step of approximation, it appears to work well in practice and the vector parametrization of the FIM allows efficient inversion via the Sherman-Morrison formula, leading to sizeable speedups, evaluated on real world architectures.

**Summary Of The Review:**

I think this is a strong paper that has the potential to help bring 2nd order optimisation to the masses, especially for industrial use cases where it is not about individual percentages on benchmarks but about tradeoffs, and the 50 epoch accuracy of Eva seems comparable to the 200 epoch accuracy of SGD, meaing it actually achieves real world speedups (especially for WRN).

---

> ### Author Response · Authors · 2022-11-07
> **Response to Reviewer p2Vi**
>
> We appreciate the positive and valuable comments.
>
> As we discussed in Section 4.5, It indeed lacks a solid theoretical analysis for both Eva and K-FAC with empirical FIM.
>
> Regarding the bonfereonni correction mentioned, we are not exactly sure how that can be applied. As our evaluations were mainly focusing on performances in terms of memory consumption, iteration time, and convergence and test accuracy.

---

> > ### Comment · Reviewer_p2Vi · 2022-11-07
> > **On bonferroni**
> >
> > When comparing test accuracy in particular, but also consumption, iteration time etc., we are comparing distributions (or at least we should be when running multiple independent seeds), so you should *ideally* perform a statistical significance test (e.g. pairwise t-test )to verify that the differences you observe are actually likely to be real vs. just "happy outliers". This won't matter as much for memory consumption etc. as I assume the differences are quite stable, but for test accuracy, especially when differences are small it can matter.  This blog post gives a more detailed overview: https://towardsdatascience.com/anova-vs-bonferroni-correction-c8573936a64e?gi=694bfff5c613

---

> > > ### Author Response · Authors · 2022-11-07
> > > **We have added pairwise t-test with Bonfereonni correction**
> > >
> > > Thank you for your kind guidance. In addition to Table 2, we perform pairwise t-test with Bonfereonni correction for the accuracy metric among SGD, K-FAC, and Eva algorithms. The results show that Eva performs closely to K-FAC in all cases (i.e., they have no significant difference with p > 0.05). K-FAC and Eva significantly outperform SGD with p < 0.05 in many cases (12 out of 18 cases), including all models with 50 epochs, VGG-19 and ResNet-110 with 100 epochs, and VGG-19 with 200 epochs on two datasets. K-FAC and Eva are slightly better than SGD in other cases while SGD is trained with sufficient epochs and/or extra tricks. We have included these summaries in Appendix C.2.

---

### Official Review · Reviewer_2rcV · 2022-10-24

**Confidence:** 3
**Correctness:** 3
**Technical Novelty And Significance:** 4
**Empirical Novelty And Significance:** 3
**Recommendation:** 6

**Clarity, Quality, Novelty And Reproducibility:**

The paper is clearly written and the proposed method is novel. For reproducibility, the method is not complicated and it seems that the results could be achieved by readers easily.

**Strength And Weaknesses:**

strength
+ it is a good try to further reduce the computational and storage burden for second-order method for training DNNs.

+ the method is not complicated and has shown great advantage in numerical experiments

weaknesses
-  there are two modifications. One is to use a mini-batch and the other is to use Sherman-Morrison formula. It is not clear how they two can coordinate each other and to achieve very good performance, while only one of them does not work.

- the authors have conducted experiments on ImageNet, however, for the most important comparison on test accuracy, the results on ImageNet are missing.

- one question is about the learning rate. I saw still Eva needs a delicate rate schedule but in theory , second-order methods are less sensitive to learning rate. Did Eva demonstrate advantages on this aspect?

**Summary Of The Paper:**

In this paper, a novel second-order technique is proposed for deep neural networks training.

**Summary Of The Review:**

Overall, it is a good try for a cheap second-order method for deep neural network training. I overall give a positive score and if the author could give convincing discussion on the currently weak points, I would like to rise my score.

---

> ### Author Response · Authors · 2022-11-07
> **Response to Reviewer 2rcV**
>
> **Question 1**: There are two modifications. One is to use a mini-batch and the other is to use Sherman-Morrison formula. It is not clear how they two can coordinate each other and to achieve very good performance, while only one of them does not work.
>
> **Response**: In Eva, we construct Kronecker vectors over a mini-batch, and it makes the second-order information a rank-one matrix, and then Sherman-Morrison is applied onto the rank-one matrix to calculate the inverse result very efficiently. Without using the Sherman-Morrison formula, the algorithm also works but requires very expensive explicit matrix inversion. Without using Kronecker vectors, one cannot use the Sherman-Morrison formula, but has to use the Woodbury formula to invert the Kronecker factors, which however needs explicit matrix inversion in the dimension of batch size [1].
>
> **Question 2**: The authors have conducted experiments on ImageNet, however, for the most important comparison on test accuracy, the results on ImageNet are missing.
>
> **Response**: In addition to Figure 3(d), the median test accuracy of the final 5 epochs on ImageNet is 76.02%, 76.25%, 76.06%, and 75.96% for SGD, Shampoo, K-FAC, and Eva, with 100, 60, 55, and 55 epochs, respectively, in training ResNet-50 whose target accuracy is 75.9% according to MLPerf. Note that Shampoo achieves slightly higher test accuracy than K-FAC as it takes more epochs, otherwise, it does not reach the target accuracy using the same settings as in K-FAC. While Figure 3(d) shows such a comparison, we have added these numerical numbers in the revised manuscript to make it more accessible.
>
> **Question 3**: One question is about the learning rate. I saw still Eva needs a delicate rate schedule but in theory, second-order methods are less sensitive to learning rate. Did Eva demonstrate advantages on this aspect?
>
> **Response**: It is true that K-FAC and Eva are less sensitive to learning rate than first-order SGD, yet they still need a reasonably good learning rate schedule for better performance. As shown in Figure 4(a), second-order methods K-FAC and Eva are less sensitive to the choice of the base learning rate than SGD. In our experiments, Eva can simply adopt the same learning rate value and learning rate schedule with SGD without further hyper-parameter tuning to achieve good convergence performance.
>
> [1] Zedong Tang, Fenlong Jiang, Maoguo Gong, Hao Li, Yue Wu, Fan Yu, Zidong Wang and Min Wang. “SKFAC: Training Neural Networks with Faster Kronecker-Factored Approximate Curvature.” CVPR, 2021.

---

### Author Response · Authors · 2022-11-07
**We have uploaded a revised manuscript [updated on 7 Nov]**

Dear reviewers:

Thank you very much for reviewing our paper and providing constructive comments. We have revised our paper following the suggestions and comments from all the reviewers. These changes are highlighted in blue in the revised manuscript. The major changes are summarized as follows:
* we compared the performance between K-FAC and Eva under different learning rate and batch size settings, in Figure 4(a)(b), and Figure 9(a)(b) (as suggested by the reviewer 2rcV and reviewer j8t4)
* we have conducted experiments for the autoencoder task on three different datasets (MNIST, Faces, and Curves), in Figure 5(a)-(f) (as suggested by the reviewer 1mYD)
* we have studied the effects of update interval for both K-FAC and Shampoo, in Figure 8(a)-(d) (as suggested by the reviewer 1mYD)

---

### Public Comment · ~Rohan_Anil1 · 2022-11-16
**This work's ImageNet results are worse than established baselines for Shampoo**

I would like to point out that this paper states results for Shampoo on ImageNet + ResNet50 are worse than established baselines from MLPerf Training 1.0 Open Submission as well from Table 1, Pg 11: https://arxiv.org/pdf/2002.09018.pdf

Batch size: 32768 requires 44 epochs or 1729 steps to achieve 75.9%
See: https://github.com/mlcommons/training_results_v1.0/blob/master/Google/results/tpu-v4-256-JAX-Distributed-Shampoo/resnet/result_0.txt

Batch size: 65536 requires 60 epochs or 1178 steps see: https://github.com/mlcommons/training_results_v1.0/blob/master/Google/benchmarks/resnet/implementations/resnet-research-JAX-Distributed-Shampoo-tpu-v4-256/train.py#L558

This work makes use of batch size of 2048 (64 x 32) for which higher order methods only require much fewer epochs, and existing results of Shampoo are better than results presented for Eva which takes 46 epochs for 3072 (96 x 32) batch size.

---

> ### Author Response · Authors · 2022-11-17
> **Update on Shampoo's configuration**
>
> **Response**: Thank you for your good reference for a well-established baseline for Shampoo on ImageNet. We are doing our best to reproduce it according to your provided configuration, and we will update the result in the revised paper if the result is reproduced in our environment. Before that, we’d like to clarify that the performance difference was mainly caused by hyper-parameter settings. In our work, we adopt similar hyper-parameters, including using the multi-step learning rate schedule, from [1] to K-FAC, Shampoo, and Eva, for a fair comparison. It is true that the convergence result for them could be further improved if using dedicated hyper-parameter tuning. For instance, Shampoo can achieve 75.9% with 44 epochs rather than 55 epochs as you mentioned. Assuming that Shampoo achieves 75.9% with 44 epochs, its end-to-end training wall-clock time is still about 1.48x longer than Eva due to Shampoo's high computational cost in its second-order update, in our hardware setting. Our code for fairly comparing different optimizers has been submitted along with the manuscript. Thanks for your comment again.
>
> [1] J Gregory Pauloski, Zhao Zhang, Lei Huang, Weijia Xu, and Ian T Foster. Convolutional neural network training with distributed K-FAC. SC, 2020.
>
>
> **-----updated on 19 Nov-----**
>
>
> **Update**: We are sorry to report that we can’t reproduce the result of Shampoo on ImageNet according to your configuration from MLPerf. The time is limited to us, and each run will take us about 6 hours in our hardware setting. We’d like to discuss this with you in the future. But before that, let’s share our efforts on tuning Shampoo.
>
> First, we can’t simply run the training script from MLPerf, which was implemented on JAX with TPUs (and relied on Google’s internal library). So we adopt the suggested hyper-parameters and attempt to reproduce it with PyTorch’s implementation from [2], which however caused training divergence.
>
> Second, we have to dive into the effects of these hyper-parameters one by one. (One limitation of Shampoo is that it has so many hyper-parameters, but the effects of them have not been well studied, causing inconvenience for tuning them). Specifically, the suggested hyper-parameters from [2] include:
>
>     learning_rate=13.0, beta1=0.95, beta2=0.85, weight_decay=0.0001, graft_type=<LayerwiseGrafting.SGD: 1>, nesterov=True, exponent_override=4.0, global_batch_size=32768 (with gradient accumulation of 16), epochs=44, and so on.
>
> We have tried several revised configurations: (1) we reduce the global batch size to 64 * 32=2048 (no gradient accumulation), as a smaller batch size typically converges easier. However, it helps training in the beginning but it can’t avoid divergence in the end; (2) we disable exponent override to avoid divergence, but it can only achieve 63% with 44 epochs (or 64.5% with 55 epochs); (3) we replace polynomial learning rate schedule by K-FAC’s multi-step learning rate schedule, and it can achieve 75% with 55 epochs, which is still under the target of 75.9%; (4) we further tune the learning rate, beta1, and beta2, but they all fail to achieve 75.9% with 55 epochs.
>
> In summary, while Shampoo has the potential of improved convergence performance, it is not easy to tune hyper-parameters successfully. In fact, the original result in our paper turned out to be the best one (achieving 75.9% with 55 epochs) in all our experimental configurations. Assuming that Shampoo can achieve 75.9% with 44 epochs, its end-to-end training time is still about 1.48x longer than Eva, which does not change our experimental conclusion.
>
> Therefore, we believe that our experimental results for Shampoo on ImageNet is good enough, and further improving it will require many efforts on tuning hyper-parameters. But the purpose of this paper is not to break the MLPerf record. As other reviewers mentioned, our experiments are well designed to support the efficiency and effectiveness of the proposed Eva algorithm. Much appreciated if you could provide a codebase that can be reproduced on a moderate GPU cluster (e.g., 32-64 GPUs) and much welcomed if you would like to have a more detailed discussion.
>
> [2] Distributed Shampoo Implementation, https://github.com/google-research/google-research/tree/master/scalable_shampoo

---

### Decision · Program_Chairs · 2023-01-20

**Decision:**

Accept: poster

**Justification For Why Not Higher Score:**

Reviewers are not strongly positive about the paper.

**Justification For Why Not Lower Score:**

A majority of reviewers vote for acceptance and the only reviewer tending slightly towards rejection did not make very strong points.

**Metareview: Summary, Strengths And Weaknesses:**

Summary:

This study proposes Eva, a second-order optimization method for training deep neural networks that approximates the Kronecker factor of K-FAC using empirical FIM with a rank one matrix. Averaging batch activations or errors construct the rank one matrix. Experimental results on various computer vision tasks show that Eva achieves similar convergence (number of steps vs. accuracy) as K-FAC, Shampoo, and M-FAC. Since Eva's computational cost per step is considerably smaller than other second-order optimization methods and is close to SGDâs cost, Eva achieves relatively fast training times in wall-clock time.

Strengths:

- it is a good try to further reduce the computational and storage burden for second-order method for training DNNs.

- the method is not complicated and has shown great advantage in numerical experiments

- the explanation of related work, background etc. is well done.

- the method is conceptually simple, but clever. I think it is non-obvious that the OPAG approximation would work as well as the K-FAC approximation a priori, and the exploitation of Sherman-Morrison and the rescaling to gradient norm in order to make the method drop-in are nice touches

- the experimental evaluation is top notch, using the related work and real world datasets and architectures

- the limitations section is an actual, honest self assessment

- the authors state most weaknessses in the related work

- The paper clearly describes the literature, the differences between the proposed and existing methods, and the experimental details, which are easy to follow.
- The proposed Eva method is straightforward (in a good sense). It boosts the practicality of second-order optimization in deep learning regarding computational and memory cost and ease of implementation and integration (as long as the fast convergence holds for other tasks). Thanks to the low computational cost, there is no need to use stale second-order information or to tune the second-order information update interval.

- Experiments are well designed to validate the effectiveness of second-order optimization methods.

- Hyperparameters specific to second-order optimization are explored. Since damping, second-order information update interval, and running average coefficient largely affect the convergence of mini-batch second-order optimization, a comparison with various values of them is valuable in verifying Evaâs robustness.

Weaknesses:

- No many since the authors have addressed most in their rebuttal.

- there is no theoretical analysis in this study, and the effectiveness of Eva is supported only by the experimental results

- The approximations made in the algorithm are not motivated in the paper.

Decision:

A majority of reviewers vote for acceptance. I, therefore, recommend accepting the paper and encourage the authors to use the feedback provided to improve the paper for its camera-ready version.

**Note From Pc:**

if the above contains the word "oral" or "spotlight" please see: "oral" presentation means -> notable-top-5% and "spotlight" means -> notable-top-25%. As stated in our emails, we are disassociating presentation type from AC recommendations